# The Arrestin-like Protein palF Contributes to Growth, Sporulation, Spore Germination, Osmolarity, and Pathogenicity of *Coniella vitis*

**DOI:** 10.3390/jof10070508

**Published:** 2024-07-22

**Authors:** Xiangtian Yin, Zihe Chen, Tinggang Li, Qibao Liu, Xilong Jiang, Xing Han, Chundong Wang, Yanfeng Wei, Lifang Yuan

**Affiliations:** 1Shandong Academy of Grape, Shandong Academy of Agricultural Sciences, Jinan 250100, China; yxt1985@163.com (X.Y.); litinggang@saas.ac.cn (T.L.); liuqibao@saas.ac.cn (Q.L.); jiangxilong@saas.ac.cn (X.J.); hanxing@nwafu.edu.cn (X.H.); cdwang1221@163.com (C.W.); weiyanfeng@saas.ac.cn (Y.W.); 2School of Landscape and Ecological Engineering, Hebei University of Engineering, Handan 056038, China; chenzihe0409@163.com

**Keywords:** *Coniella vitis*, *palF*, RNA interference, growth, virulence

## Abstract

*Coniella vitis* is a dominant phytopathogen of grape white rot in China, significantly impacting grape yield and quality. Previous studies showed that the growth and pathogenicity of *C. vitis* were affected by the environmental pH. Arrestin-like protein PalF plays a key role in mediating the activation of an intracellular-signaling cascade in response to alkaline ambient. However, it remains unclear whether palF affects the growth, development, and virulence of *C. vitis* during the sensing of environmental pH changes. In this study, we identified a homologous gene of PalF/Rim8 in *C. vitis* and constructed *CvpalF*-silenced strains via RNA interference. *CvpalF*-silenced strains exhibited impaired fungal growth at neutral/alkaline pH, accompanied by reduced pathogenicity compared to the wild-type (WT) and empty vector control (CK) strains. The distance between the hyphal branches was significantly increased in the *CvpalF*-silenced strains. Additionally, *CvpalF*-silenced strains showed increased sensitivity to NaCl, H_2_O_2_, and Congo red, and decreased sensitive to CaSO_4_. RT-qPCR analysis demonstrated that the expression level of genes related to pectinase and cellulase were significantly down-regulated in *CvpalF*-silenced strains compared to WT and CK strains. Moreover, the expression of *PacC*, *PalA/B/C/F/H/I* was directly or indirectly affected by silencing *CvpalF*. Additionally, the expression of genes related to plant cell wall-degrading enzymes, which are key virulence factors for plant pathogenic fungi, was regulated by *CvpalF*. Our results indicate the important roles of *CvpalF* in growth, osmotolerance, and pathogenicity in *C. vitis*.

## 1. Introduction

Grape white rot is a destructive disease in grape-growing regions worldwide, particularly in warm and rainy climates, leading to 20–30% production loss and 30–50% economic loss [1,2,3]. The phytopathogens responsible for the disease include *Coniella diplodiella*, *C. diplodiopsis*, *C. fragariae*, *Pilidiella castaneicola,* and *C. vitis* [1,4,5]. *C. vitis* has been reported as the dominant phytopathogen of grape white rot in China, causing typical symptoms on fruit, shoot, and leaves. Diseased fruit and leaves are covered with dense and black pycnidia [6,7,8]. Previous research has highlighted the acidic environment as a key aspect of the pathogenicity of *C. vitis*, with the ability to produce various organic acids and cell wall-degrading enzymes, which are crucial for its virulence [9].

Environmental pH is a key factor that influences the growth, development, secondary metabolism, and virulence of the phytopathogen. Fungal plant pathogens can successfully identify and invade hosts by responding quickly and effectively to environmental pH changes. The pathogenicity of *Fusarium oxysporum* on strawberry roots is higher in acidic soil than in neutral soil [10]. *Penicillium expansum* enhances its colonization ability on apples by sensing organic acids secreted by apples [11]. *Colletotrichum gloeosporioides* can increase its pathogenicity to avocados by secreting ammonia to alkalize the surrounding environment [12]. *C. vitis* exhibits rapid growth and heightened virulence in acidic environments, particularly around pH 3, compared to neutral or alkaline conditions [9]. Additionally, it is notable that the pH of table and wine grape leaves and fruits ranges from 2.79 to 3.77 [9,13]. Therefore, it is reasonable to speculate that pH is an important factor in the infection of grapes by *C. vitis*.

Phytopathogens have evolved various mechanisms to adapt to diverse environmental conditions, including the Pal/Rim–pH pathway. This pathway comprises several key proteins, including PalA/Rim20, PalB/Rim13, PalC/Rim23, PalF/Rim8, PalH/Rim21, PalI/Rim9, and the transcription factor PacC/Rim101 [14,15]. This Pal/Rim–pH pathway plays a crucial role in sensing and responding to changes in environmental pH. PacC modulates a wide range of phenotypes, including mycelial growth, spore germination, sporulation, and virulence. The loss of *PePacC* leads to a noticeable decrease in growth and conidiation of *P. expansum* and *Trichothecium roseum*, as well as reduced pathogenicity [16,17]. Under neutral and alkaline conditions, the ambient pH signal is sensed by PalH, which subsequently facilitates the ubiquitination of the arrestin PalF [18,19,20]. Ubiquitinated PalF triggers and activates the proteolysis of PacC/Rim101 in response to alkaline pH, resulting in the generation of PacC27 (kDa) by two successive proteolytic cleavages from PacC72 (kDa). PacC27 then localizes to the nucleus to modulate gene expression [19,21]. PalF/Rim8 can promote Pal/Rim–pH pathway activation in *Saccharomyces cerevisiae*, *Aspergillus nidulans*, and *Candida albicans*, but there are significant differences in PalF activity among these model organisms [22,23]. For example, Rim8 ubiquitination is pH-independent in *S. cerevisiae* but pH-dependent in *A. nidulans* [24]. However, it remains unclear whether palF affects the growth, development, and virulence of fungi in response to changes in environmental pH. Although the interaction between PalF and the C-terminal cytoplasmic tail of PalH highlights the role of PalF as a mediator of pH signaling within the cell, the role of PalF in the growth, development, and virulence of fungi needs further investigation.

Previous research identified CvpalF in *C. vitis* as a homologue of PalF/Rim8, with its expression down-regulated at alkaline pH [9]. The present study (i) compares the sequence identity and genetic relationship between *CvpalF* and those of other pathogenic fungi; (ii) silences the *CvpalF* gene via RNA interference; (iii) elucidates the effects of *CvpalF* on growth, development, and stress response under different pH conditions and pathogenicity; and (iv) demonstrates the regulation of *CvpalF* on genes in Pal-signaling pathways and other *C. vitis* virulence genes. These findings provide new insights into the pathogenic mechanism of *C. vitis* regulated by pH.

## 2. Materials and Methods

### 2.1. Strains, Culture Conditions, DNA Extraction, and Bioinformatics Analysis

*C. vitis* GP1 (strain no. 23888) is a highly virulent strain causing grape white rot, and presently preserved in the China General Microbiological Culture Collection Center [7]. The wild-type strain *C. vitis* GP1 and transformants generated in this study were cultured on potato dextrose agar (PDA, Difco, Detroit, MI, USA, pH = 6.5) plates at 28 °C in the dark for 3 days and potato dextrose broth (PDB) media at 28 °C, in the dark, and at 180 rpm.

For genomic DNA extraction, the fungal strains were grown on PDA (28 °C, darkness) for 3 days on the surface of cellophane membranes inoculated with 4-mm mycelial plugs. Genomic DNA was extracted with the DNeasy Plant Mini Kit (Qiagen, Hilden, Germany) according to the manufacturer’s instructions.

The genomic DNA of strain *C. vitis* GP1 was sequenced at Novogene Bioinformatics Technology Co., Ltd. (Nanjing, China) using a combination of third-generation PacBio and second-generation Illumina. The sequence data for *CvpalF* were extracted from the annotation results and are presented in Appendix A. Homologous sequences of PalF protein from other fungal species were downloaded from the NCBI database. Multiple sequence alignment and phylogenetic tree construction were performed using MEGA 7.0 with the maximum likelihood clustering method. Structural domains of PalF proteins were predicted using the online biology tool in NCBI (https://www.ncbi.nlm.nih.gov/Structure/cdd/wrpsb.cgi?RID=XG0R7TS6016&mode=all accessed on 2 February 2024).

### 2.2. Construction of the CvpalF-Silenced Strains

#### 2.2.1. Hygromycin Sensitivity

*C. vitis* GP1 was cultured on PDA for 3 days at 28 °C in the dark. Following this, mycelial plugs (5 mm diameter) were transferred to fresh PDA-containing hygromycin B (Roche) at concentrations of 0, 1, 5, 10, 50, 100, 200, and 300 μg/mL. The plates were incubated at 28 °C in the dark for 3 d. The diameters of the developed colonies were measured. 

#### 2.2.2. Construction of the *CvpalF*-Silencing Plasmid

pSlient-1 was used to construct the gene *CvpalF*-silencing vector. Two reverse-complementary *CvpalF*-silencing targets were amplified using the primers of palF1/2 and palF3/4 (Table 1). Firstly, the restriction endonucleases XhoI (New England Biolabs, Beijing, China) were used to digest at 37 °C and the *CvpalF* sense forward fragment was inserted into the restriction sites by one step of the Cloning kit (Vazyme, Nanjing, China). Secondly, the restriction endonucleases Kpn I (New England Biolabs, Beijing, China) was used to digest, and the *CvpalF* sense reverse-complementary fragment was inserted into the corresponding restriction site, yielding the final silencing vector p*CvpalF*i (Appendix A). Subsequently, the resultant constructs were introduced into *E. coli* DH5α, and positive clones were chosen for PCR identification with primers LFcheck 1F/R, LFcheck 2F/R (Table 1), and sequencing to ensure the validation of their accuracy.

#### 2.2.3. Transformation Procedure

The spore of *C. vitis* GP1 was cultured in the yeast extract peptone dextrose (YEPD) media (with 10% grape juice) at 28 ℃ in the dark for 20 h with 180 rpm for preparation of protoplast according to the method described by Zou [25]. The silencing vector p*CvpalF*i was transferred into the *C. vitis* GP1 strain by PEG-mediated protoplast transformation [26,27]. Transformants were selected and cultured on PDA supplemented with hygromycin B (200 μg/mL) for three successive subcultures. After extraction using the EZ-DNAaway RNA Mini-Preps Kit (Sangon Biotech, Shanghai, China) according to the manufacturer’s protocol, *CvpalF*-silenced strains were confirmed by PCR amplification with LFcheck 1F/R, LFcheck 2F/R, and Fhyg/Rhyg primers (Table 1). The PCR cycling conditions were as follows: initial denaturation at 95 °C for 3 min; 34 cycles of denaturation at 95 °C for 30 s, annealing at 58 °C for 30 s, and extension at 72 °C for 1 min; followed by a final extension step at 72 °C for 10 min.

#### 2.2.4. Gene-Expression Analysis

The wild-type (WT) strain *C. vitis* GP1, vector control (CK) strain, and *CvpalF* transformants *CvpalF*i-17 and *CvpalF*i-29 were cultured in PDB media of pH 3, 5, 7, and 9 for 3 days at 28 °C, in the dark, 180 rpm. The pH of the PDA medium was adjusted using 0.2 M Na_2_HPO_4_·12H_2_O, 0.1 M C_6_H_8_O·7H_2_O and NaOH as described previously [9]. Total RNA was extracted using EZ-10 DNAaway RNA Mini-Preps Kit (Sangon Biotech, Shanghai, China) according to the protocol. RNA samples (1 μg) were reverse transcribed using the HiScript III 1st Strand cDNA Synthesis Kit (+gDNA wiper, Vazyme, Nanjing, China). The primers of *palF* gene designed for quantitative real-time PCR (qRT-PCR) were listed in Appendix A. qRT-PCR analysis was performed in a two-step method using Taq Pro Universal SYBR qPCR Master Mix (Vazyme, Nanjing, China) in a CFX-96 real-time PCR detection system (Bio-Rad, Hercules, CA, USA). The qRT-PCR mixture (20 μL) consisted of 10 μL of 2 × qPCR Master Mix, 0.4 μL of each primer (10 μM), and 2 μL of cDNA. The qRT-PCR conditions were as follows: 95 °C for 30 s, followed by 40 cycles of 95 °C for 3 s and 60 °C for 20 s. The actin gene was used as the reference gene and relative expression levels were measured using the 2^(−ΔΔCt)^ analysis method [28]. All the experiments were performed with three technical replicates of three biological replicates.

### 2.3. Silenced Strains Characterization

#### 2.3.1. Growth under Different pH

##### On Solid Media

Strains WT, CK, and *CvpalF* transformants *CvpalFi*-17 and *CvpalFi*-29 were cultured on PDA and amended to pH 3, 5, 7, and 9. The pH was adjusted as described by Yuan [9]. After 3 days of incubation (28 °C in the dark), the colony diameters were measured, and the colony edges were observed by a light microscope (Optika B-383PL, Ponteranica, Italy). The WT, CK, *CvpalF*i-17, and *CvpalF*i-29 strains were grown on PDA supplemented with a pH indicator (0.005% Bromothymol blue) at 28 °C, in the dark, after 3 days to observe the color of the PDA. The experiment was performed with three biological replicates and repeated three times.

##### In Liquid Media

Spore suspension of strains WT, CK, and *CvpalF* transformants (*CvpalFi*-17 and *CvpalFi*-29) was prepared as described previously. PDB (potato dextrose broth) (100 mL, pH = 6.5) inoculated with 1 mL spore suspension (1 × 10^6^ spore/mL) was cultured at 28 °C, darkness, at 180 rpm. After 3 days, the mycelia were harvested by filtering through four layers of cheesecloth, dried at 60 °C, and weighted with an analytical balance to determine the mycelia biomass. The pH of each cuture was measured every day with a pH meter (Sartorius, PB-10).

#### 2.3.2. Spore Production and Vitality

Strains WT, CK, and *CvpalF* transformants (*CvpalFi*-17 and *CvpalFi*-29) were cultured on PDA at pH3, 5, 7, and 9 for 14 days at 28 °C in the dark. Spores were collected by washing each colony surface with 40 mL of 0.05% (*v*/*v*) Tween 80 (Sigma-Aldrich, Copenhagen, Denmark) in distilled water. Suspensions were filtered on miracloth (Calbiochem) into 50 mL tubes. The spore concentrations were determined by hemocytometer, and spore morphology was observed under light microscopy (Optika B-383PL, Ponteranica, Italy). Spore vitality was verified as proposed by Yuan et al. [9].

#### 2.3.3. Pathogenicity

Mycelial plugs (5 mm in diameter) taken from 3-day-old *C. vitis* strains WT, CK, and *CvpalF* transformant (*CvpalFi*-17 and *CvpalFi*-29) colonies on PDA were used to inoculate detached grape leaves (cv. Red Globe). Detached grape leaves inoculated with PDA plugs served as a control. Inoculated leaves were incubated in a greenhouse at 28 °C, 90% RH, and under natural photoperiod. The spot diameters on the leaves were determined at 1, 2, and 3 dpi (days post inoculation). The reduced rate = [the lesion diameter of the WT strain—the lesion diameter of transformants]/the lesion diameter of WT strain × 100%). The experiment was repeated three times with three biological replicates.

#### 2.3.4. Effects of Osmotic Stress

The effects of *CvPalF* on osmotic stress adaptability on PDA amended with KCl (1 M), NaCl (1 M), CaSO_4_ (1 M), MgSO_4_ (1 M), H_2_O_2_ (1 M), or Congo red (1 M). Plates were inoculated with a mycelial plug (5 mm in diameter) cut from 3-day-old *C. vitis* strains WT, CK, *CvpalFi*-17, and *CvpalFi*-29 cultures on PDA (28 °C, in the dark). After 7 days of incubation (28 ℃, in the dark) the colony diameter was measured. Inhibition rate = [the diameter of the WT strain—the diameter of transformants]/the diameter of WT strain × 100%.

#### 2.3.5. Virlence Gene Expression

Conidia of *C. vitis* strains WT, CK, *CvpalFi*-17, and *CvpalFi*-29 were inoculated in 20 mL of PDB with a pH of 3 to obtain a final concentration of 3 × 10^6^ spores/mL. Flasks were incubated at 28 °C, in the dark, with shaking at 180 rpm for 24 h. Then, the cultures were shifted to PDB with pH values of 3 and pH 9 for further 15, 30, 60, or 120 min according to the method of Hervas–Aguilar et al. [29]. Subsequently, mycelia were harvested using four layers of cheesecloth. Total RNA was extracted and conducted as described by Yuan et al. [9] and Mushtaq et al. [30]. qRT-PCR was conducted using the primer pairs of the Pal-signaling pathway and other virulence genes listed in Appendix A, and the rection conditions were the same as described in Section 2.2.4 in this study.

### 2.4. Statistical Analysis

All data in this study were analyzed by SPSS Version 22.0, and one-way ANOVA with Tukey’s test was used for multiple comparison analyses. Significance was accepted at *p* < 0.05 [31,32]. Each experiment was performed with three biological replicates and repeated three times, and the data were presented as averages ± SE.

## 3. Results

### 3.1. Bioinformatics Analysis of CvpalF

The *CvpalF* gene in the *C. vitis* GP1 genome was 2421 bp long and encoded a total of 806 amino acids (Appendix A). The SMART structural domain analysis showed that CvpalF protein contains arrenstin-N and arrestin-C domains located at Positions 49–204 and 306–458, respectively (Figure 1A). These domains showed high conservation in *Aspergillus nidulatus*, *Pyricularia oryzae,* and *Neurospra crassa* (Figure 1B). The phylogenetic tree of PalF proteins (Figure 1C) showed that *CvPalF* had a relatively close evolutionary relationship with *P. oryzae* and *N. crassa* but a distant relationship with *A. nidulatus*, *A. fumigatus,* and *Penicillium diatomitis*.

### 3.2. Construction of CvpalF-Silenced Strains

The *pCvpalF*i vector and the pSlient-1 vector used as a vacant vector control (Figure 2A, Appendix A) were inserted via PEG-mediated protoplast transformation into WT strain *C. vitis* GP1. Hygromycin B fully suppressed the growth of *C. vitis* GP1 at 100 μg/mL (Appendix A). To avoid false positives, Hygromycin B at 200 μg/mL was used for selecting transformants. Six transformants were randomly selected. PCR amplifications with primers LFcheck 1F/1R, LFcheck 2F/2R, and F/Rhyg confirm transformation, yielding bands of 1072, 1164, and 750 bp, respectively (Figure 2B). The expression level of the *CvpalF* gene was significantly reduced in the *CvpalF*i-17 and *CvpalF*i-29 transformants, with a gene-silencing efficiency of 76.17–98.93% (Figure 2C,D). Meanwhile, the expression level of the *CvpalF* gene was not significantly changed in CK and WT strains.

### 3.3. Silenced Strains Characterization

#### 3.3.1. The Growth of *CvpalF*-Silenced Strains Under Different pH Values

*CvpalF*-silenced strains *CvpalF*i-17 and *CvpalF*i-29 exhibited impaired growth compared with WT and CK strains, with significant inhibition under alkaline conditions (Figure 3A). Compared to WT and CK strains, the growth inhibition (Figure 3B) of the two *CvpalF* silenced strains ranged from 28.48 to 46.81%. Additionally, the colony edges appeared shorter and sparser in *CvpalF*i-17 and *CvpalF*i-29 (Figure 3C).

The mycelial biomass of strains *CvpalF*i-17 and *CvpalF*i-29 was significantly reduced by the pH of PDB (Figure 3D). The mycelial biomasses of *CvpalF*-silenced strains *CvpalF*i-17 and *CvpalF*i-29 declined 20.86 and 14.75%, 65.71 and 66.94%, and 52.06 and 58.22% at pH 3, 5, and 7, respectively (Figure 3D). However, the mycelial weight of *CvpalF*i-17 and *CvpalF*i-29 had no significant reduction when grown in PDB at pH 9 (Figure 3D).

#### 3.3.2. The pH Change of Liquid Media

The pH values of PDB decreased during the cultivation of WT strain, CK, and *CvpalF* silenced strains *CvpalF*i-17 and *CvpalF*i-29, with initial pH values of 3, 5, and 7 (Figure 4A–C). During cultivation with an initial pH of 7, the pH values of PDB for *CvpalF*i-17 and *CvpalF*i-29 were slightly higher than those for the WT (Figure 4C). However, there was no significant difference in the pH values of PDB between the WT and the two *CvpalF*-silenced strains when the initial pH was 9 (Figure 4D). Additionally, the WT demonstrated a stronger capacity to produce acid and decrease the environmental pH compared to *CvpalF*i-17 and *CvpalF*i-29 (Appendix A).

#### 3.3.3. Sporulation and Spore Germination of *CvpalF*-Silenced Strains under Different pH Values

Sporulation of *CvpalF*-silenced strains *CvpalF*i-17 and *CvpalF*i-29 was reduced by 32.83% and 21.64%, respectively, compared to the WT and CK strains when incubated in PDA with pH 3 at 28 °C for 14 days. In pH 5, their sporulation was reduced by 24.34% and 21.71%, respectively. Moreover, the sporulation of strains *CvpalF*i-17 and *CvpalF*i-29 was further reduced by 84.23% and 58.62%, respectively, compared to the WT and CK strains when incubated in PDA media with pH 7. Particularly in alkaline environments, the sporulation ability of transformant *CvpalF*i-17 was completely lost, while the sporulation of Strain *CvpalF*i-29 was reduced by 68.61% (Figure 5A). However, *CvpalF* did not contribute to spore germination, regardless of the environmental pH. The rate of spore germination did not significantly differ between *CvpalF*i-17, *CvpalF*i-29, CK, and WT strains (Figure 5B).

#### 3.3.4. Pathogenicity of *CvpalF*-Silenced Strains

On detached leaves of cv. Red globe, *CvpalF*-silenced strains *CvpalF*i-17 and *CvpalF*i-29, developed brown spots much slower than Strains WT and CK. Compared with WT and CK, a reduction of spot diameter by 55.75 and 32.87% in strains *CvpalF*i-17 and *CvpalF*i-29, respectively (Figure 6A,B and Appendix A). 

#### 3.3.5. The Responses of *CvpalF*-Silenced Strains to Cell Wall Integrity, Oxidation, and Ionic Stress

Under cell wall integrity of Congo red and oxidative stress induced by H_2_O_2_, the growth of *CvpalF*-silenced strains *CvpalF*i-17 and *CvpalF*i-29 were partly inhibited (Figure 7). *CvpalF*-silenced strains *CvpalF*i-17 and *CvpalF*i-29 were notably more sensitive to 1 M NaCl and exhibited approximately 45–50% greater growth inhibition than the WT and CK strains. While *CvpalF*-silenced strains *CvpalF*i-17 and *CvpalF*i-29 were less sensitive to 1 M KCl and exhibited approximately only 9.81% growth inhibition than the WT and CK strains. Interestingly, *CvpalF*-silenced strains showed decreased sensitivity to 1 M CaSO_4_, and the growth of *CvpalF*-silenced strains was faster than the WT and CK strains. In the presence of 1 M MgSO_4_, no significant differences (*p* > 0.05) were found between the *CvpalF*-silenced strains, WT, and CK strains (Figure 7 and Appendix A).

#### 3.3.6. CvpalF Affected the Expression of Genes Related to Pal-Signaling Pathway and Plant Cell Wall-Degrading Enzyme

The expression levels of *CvpalA*, *CvpalB*, *CvpalC*, *CvpalH*, *CvpalI,* and *CvpacC* in *CvpalF*-silenced strains were significantly down-regulated by approximately 1.27–6.25 folds (Figure 8A). Additionally, genes related to plant cell wall-degrading enzymes, including *CvpmeA*, *CvpmeB*, *CvpLL* (pectin lyase-like protein), *CV_V008718*, *CV_V004839*, *CV_V009031*, and *CV_V008957*, were significantly down-regulated by approximately 1.46–23.08 folds (Figure 8B). The expression level of *CvpacC* increased by 3.69–5.80 folds after 15 min when the pH shifted from 3 to 9, while the expression level of *CvpalH* increased by 6.37–6.93 folds after 30 min under the same pH shift condition (Figure 8C). In addition, the expression level of *CvpmeA* was up-regulated in *CvpalF*i-17 and *CvpalF*i-29 strains when the environment pH shifted from 3 to 9, but it was lower than that in WT. The expression level of *CvexgD* was increased by 2.09–2.17 folds in *CvpalF*i-17 and *CvpalF*i-29 strains compared to WT at 30 min in the pH shift experiment. The expression levels of *CV_V004126* and *CV_V004839* were lower than that in WT, although the expression level of *CV_V004839* increased by 0.84–0.85 folds after 30 min when pH shifted from 3 to 9 (Figure 8C).

## 4. Discussion

Grape white rot, caused by *C. vitis*, has been a major concern in grape production and the wine industry [4,33,34]. Unlike most plant pathogens, the host plants of *C. vitis* reported so far are grapes and Virginia Creeper (*Parthenocissus quinquefolia*) with a pH value of about 3 [35,36]. Previous studies show that *C. vitis* grows slowly and includes a loss or reduction of pathogenicity in neutral and alkaline environments [9]. The Pal/Rim 101 signal pathway regulates the adaption to changes in the environmental pH for the survival and development of fungi, and *PalF* was known to trigger the Pal/Rim 101 signal pathway in response to the change of the environmental pH [20]. In this study, a homologous gene of PalF/Rim8 was identified in *C. vitis*. The *CvpalF-*silenced strains *CvpalF*i-17 and *CvpalF*i-29 were also obtained using an RNAi approach with the fungal transformation vector pSilent-1 through the PEG-mediated protoplast transformation. The applied method was successful in both Ascomycota and Basidiomycota [37,38,39]. Furthermore, the role of *CvpalF* in growth, virulence, and response to the different pH conditions was elucidated. Compared with *C. vitis* WT and CK, *CvpalF*-silenced strains reduce linear growth and mycelia biomass production while worsening pathogenicity (Figure 3A and Figure 6).

Pathogenic fungi alkalize or acidify the host tissue during infection to ensure their normal growth, development, and pathogenicity [40]. PacC plays a crucial role in the Pal/Rim-signaling pathway. PacC contains three Cys2His2 zinc finger motifs. Under alkaline conditions, PacC72 undergoes a two-step proteolytic process to degrade into PacC27. PacC27 then enters the nucleus and binds to the 5′-GCCARG-3′ sequence in the promoter regions of pH-responsive genes, thereby regulating the expression of these genes [14]. *B. cinerea* secretes organic acids to acidify some of the host plant tissues during colonization, while the *BcpacC* deletion mutant revealed a decreased capacity to acidify their environment due to a decrease in the production of oxalic acid [13,41,42]. Previous research indicates that the expression of *CvpacC* is increased 2.77-fold under alkaline conditions compared to acidic conditions (unpublished), and over 200 genes related to carboxylic acid metabolism are differentially expressed [9]. In this study, we found that *C. vitis* decreased the environmental pH in vitro, and the capability of acidification of *C. vitis* in PDB media of pH 5 and pH 7 were slightly declined when *CvpalF* was silenced (Figure 4B,C). These results suggested that *CvpalF* sensed the environmental pH but cannot directly regulate the metabolism of organic acids to change the environmental pH. Whether *CvpacC* is responsible for the ability to secrete organic acids to acidify the environmental pH needs to be verified further.

Previous studies have shown *pacC* and five pal genes, excluding *palI,* have proved essential for cellular response to osmotic salts/agents, and their osmoregulative roles are all pH-dependent in many fungi [43,44]. *PacC*-silenced strains of *Ganoderma lucidum* were more sensitive to 0.3 M NaCl and KCl [45]. *Fusarium oxysporum* is an important and diverse soilborne plant pathogen infecting almost 150 plant species while *FopacC* transformants increased sensitivity to NaCl [46,47]. *Fusarium graminearum* (Fg), a destructive fungal pathogen of cereal crops, and *FgPacC*, which can directly bind and inhibit the HAT activity of *FgGcn5*, subsequently enhance the adaption of *F. graminearum* to host-derived high-iron stress during infection [26]. In this study, we found that *CvpalF*i-17 and *CvpalF*i-29 showed defects in cell walls and were more sensitive to 1 M NaCl compared to WT and CK strains. Our results suggested that *palF* in the Pal signal pathway was required for osmotolerance adaptation. The mechanisms that respond to ionic pressure in the environment in *C. vitis* need further elucidation, and whether *CvPacC* is involved in this process and its role remains to be explained. Additionally, reactive oxygen species burst is an early defense response in most plant hosts during the infection of pathogenic fungi [48,49]. *CvpalF*-silenced strains exhibited increased sensitivity to H_2_O_2_, which might contribute to decline in the virulence of grapes.

Plant cell walls serve as the first line of defense against pathogen infection [50]. Polygalacturonases play an important role in cell wall degrading, and pectin methylesterase (PME) contributes to this progress [51]. Previous studies have shown that the ability of pal/rim mutants to secrete extracellular enzymes is affected. For example, the expression levels of genes encoding chitin synthases (CHS) and glucan synthase (GLS) were reduced in the *palB* mutant of *Ustilago maydis*, which was accompanied by changes in the cell wall structure and chemical composition [52]. In *Botrytis cinerea* and *F. graminearum*, *Bcpme1* and *FgPme1* mutants showed lower virulence on host plant tissues, such as apple and grapevine for *B. cinerea* and wheat for *F. graminearum* [53,54]. In this study, we found that silencing *CvpalF* resulted in the decline of the expression levels of genes related to pectinase and cellulase (e.g., genes *CvpmeA* and *CvpmeB*) (Figure 7B). The results suggested that *CvpalF* is involved in regulating the production of cell wall-degrading enzymes in *C. vitis*, which is important for its virulence.

## 5. Conclusions

In this study, the role of arrestin-like protein CvpalF was revealed by constructing *CvpalF*-silenced strains via RNA interference. Our results suggested that *CvpalF* is required for the growth and sporulation of *C. vitis* in neutral/alkaline pH conditions. *CvpalF*-silenced strains exhibited impaired fungal growth and decreased sporulation capacity compared to the WT and CK strains. *CvpalF* is essential for virulence and adaptability to environmental stresses, such as osmotic stress, oxidative stress, and cell wall stress. In addition, the expression of *PacC*, *PalA/B/C/F/H/I* was directly or indirectly regulated by *CvpalF*. These results contribute to a better understanding of the mechanism underlying Pal pathway regulation in *C. vitis*.

## Figures and Tables

**Figure 1 jof-10-00508-f001:**
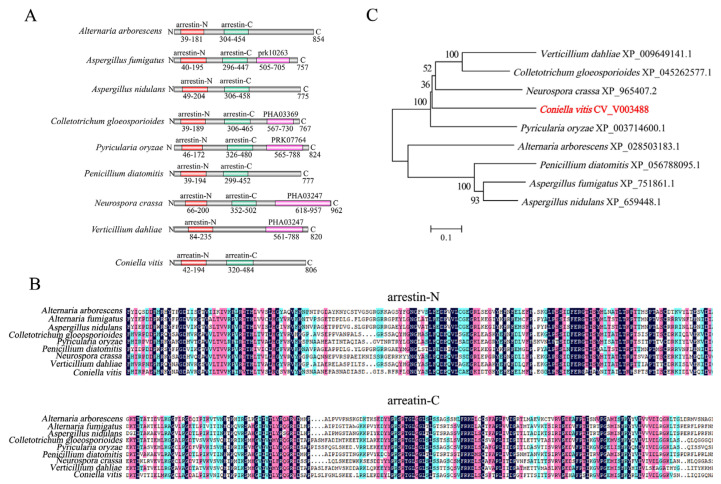
Bioinformatic analysis of *palF* genes in different fungal species: protein domain (**A**), multiple sequence alignment (**B**), and phylogenetic tree (**C**).

**Figure 2 jof-10-00508-f002:**
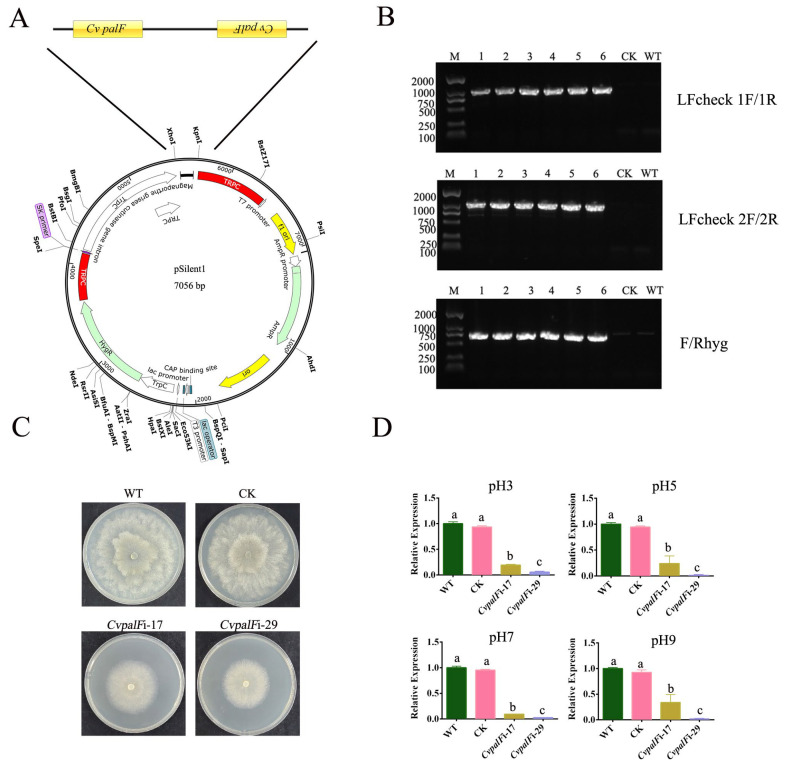
Construction and assay of *C. vitis* transformants: (**A**) Recombinant pSilent-1 vector. (**B**) PCR profile of Hygromycin (Hyg) resistance gene amplified by Primers LFcheck 1F/1R, LFcheck 2F/2R), and Fhyg/Rhyg (Table 1). Lane M: 100–2000 bp DNA marker, Lanes 1–6: different *C. vitis* transformants, CK: *C. vitis* transformant with empty vector, WT: wild type *C. vitis* strain GP1. (**C**) Colony morphology of WT, CK, and *C. vitis* transformant strains *CvpalF*i-17 and *CvpalF*i-29 after 3 days on Potato Dextrose Agar at 28 ℃ in the dark. (**D**) *Cvpal* gene expression in *C. vitis* WT, CK, *CvpalF*i-17, and *CvpalF*i-29 cropped on potato dextrose broth under different initial pH at 28 ℃, for 1 day, in the dark. For each pH tested, the *CvpalF* relative expression in the WT strain was used to normalize the values. Data are the means of three experiments ± se. In each chart, different letters indicate significant differences (*p* < 0.05) according to Tukey’s test.

**Figure 3 jof-10-00508-f003:**
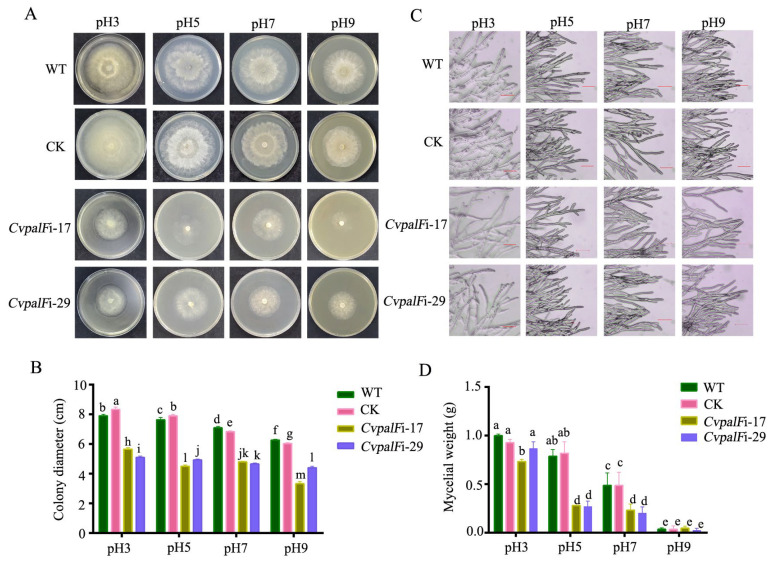
*C. vitis* GP1 (WT), GP1 empty vector transformant control (CK), and two *CvpalF* silenced strains (*CvpalFi*-17 and *CvpalFi*-29) growth on Potato Dextrose Agar (PDA) or in Potato Dextrose Broth (PDB) under pH 3, 5, 7, or 9: colony morphologies (**A**), diameter (**B**), and hyphae morphology (**C**) on PDA, mycelial biomass on PDB, scale bar = 50 μm. (**D**). For culture conditions, see the Material and Methods section. Histograms report an average of nine replicates ± SE. In each chart, different letters indicate significant differences (*p* < 0.05) according to Tukey’s test.

**Figure 4 jof-10-00508-f004:**
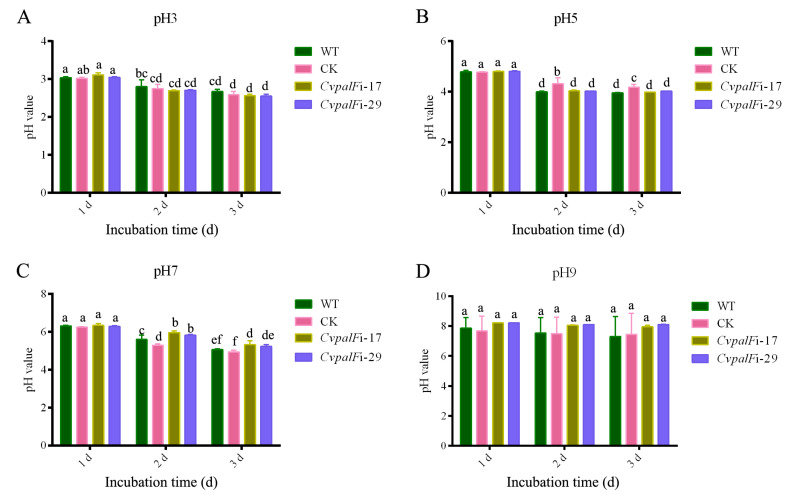
Variation of pH in Potato Dextrose Broth (PDB) during the growth of *C. vitis* GP1 (WT), GP1 empty vector transformant control (CK), and two *CvpalF*-silenced strains (*CvpalFi*-17 and *CvpalFi*-29). pH3, pH5, pH7, and pH9 indicate the initial pH of PDB. Values are the average of nine replicates ± SE. In each chart, different letters indicate significant differences according to Tukey’s test at *p* < 0.05. In each chart, different letters indicate significant differences (*p* < 0.05) according to Tukey’s test.

**Figure 5 jof-10-00508-f005:**
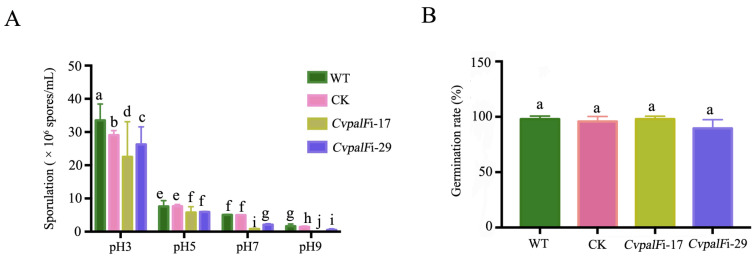
Effects of *CvpalF* on the sporulation and spore germination in *C. vitis*. (**A**) Sporulation of the *C. vitis* GP1 (WT), GP1 empty vector transformant control (CK), and two *CvpalF*-silenced strains (*CvpalFi*-17 and *CvpalFi*-29) from 14-day-old culture on Potato Dextrose Agar (PDA) at 28 °C in the dark. (**B**) Spore germination rate after 16 h of incubation (28 °C, in the dark) in 10% grape juice. Values are the average of nine replicates ± SE. In each chart, different letters indicate significant differences according to Tukey’s test at *p* < 0.05.

**Figure 6 jof-10-00508-f006:**
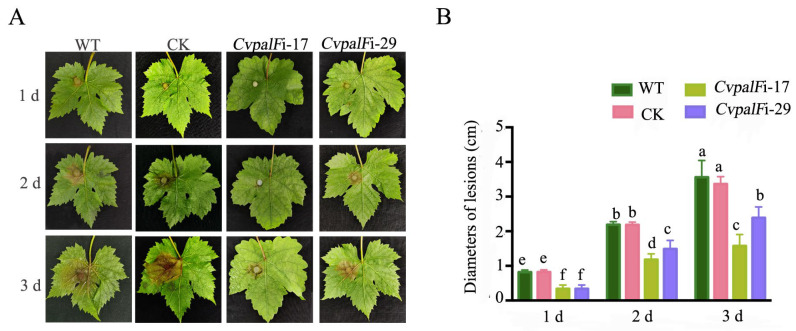
Pathogenicity test of *C. vitis* strain GP1 (WT), GP1 empty vector transformant (CK), and two *CvpalF*-silenced strains (*CvpalF*i-17 and *CvpalF*i-29) on *Vitis vinifera* cv Red globe on detached leaves: (**A**) symptoms and lesion diameters (**B**). Histograms report the average of nine replicates ± SE, different letters indicate significant differences (*p* < 0.05) according to Tukey’s test.

**Figure 7 jof-10-00508-f007:**
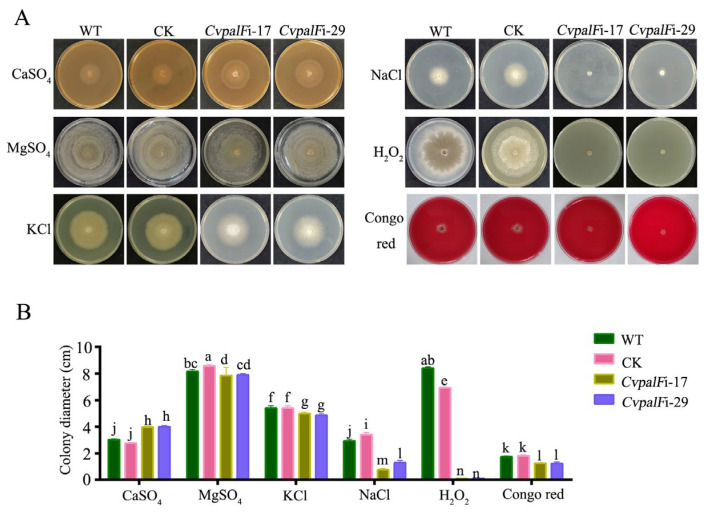
Response to different stresses of *C.vitis* GP1 (WT), GP1 empty vector transformant (CK), and two *CvpalF-*silenced strains (*CvpalF*i-17 and *CvpalF*i-29) growth for 7 days (28 ℃, in the dark) on Potato Dextrose Agar amended with 1M KCl, NaCl, CaSO_4_, MgSO_4_, H_2_O_2_, or Congo red: (**A**) colony morphologies and (**B**) colony diameter. Histograms report the average of nine replicates ± SE, different letters indicate significant differences (*p* < 0.05) according to Tukey’s test.

**Figure 8 jof-10-00508-f008:**
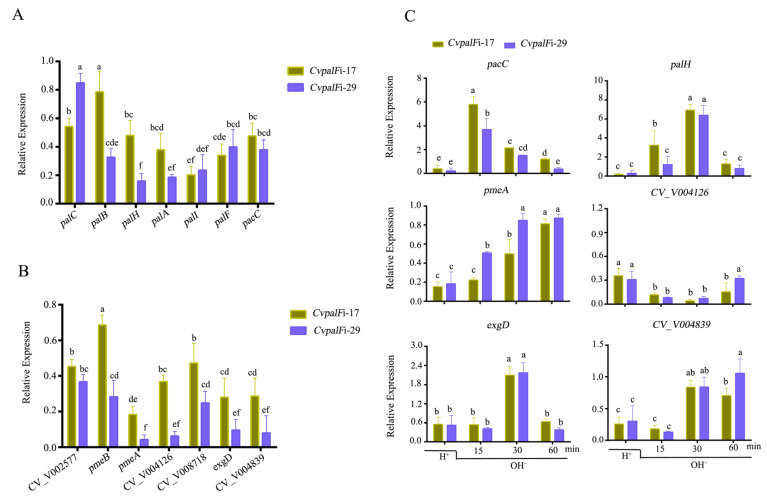
*C. vitis* GP1 (WT) and two *CvpalF-*silenced strains (*CvpalF*i-17 and *CvpalF*i-29) of gene expression: (**A**) genes related to Pal-signaling pathway, (**B**) plant cell wall-degrading enzymes, and (**C**) Virulence gene expression upon shifting from pH 3 to pH 9. Relative expression was normalized using actin as an internal control. The relative gene-expression levels in WT were taken as the value of 1. different letters indicate significant differences (*p* < 0.05) according to Tukey’s test. The primers used are listed in Appendix A.

**Table 1 jof-10-00508-t001:** Primers used in this study.

Name	Sequence (5′-3′)	Fragments (bp)
palF1	AGCATCGATACCGTCGACCACCGTTTGGAGGTTATTGTGGAC	485
palF2	AGCAAGCTTGTACGTACGAACTGCGGAGGAGGTATGTAGT	
palF3	TAAGTGGATCCGGGGCCCAGACCGTTTGGAGGTTATTGTGGAC	485
palF4	TCGCATGCTAAGGCCTGTGAACTGCGGAGGAGGTATGTAGT	
LFcheck 1F	ATGAGCAAGCGGACGGAGTG	1072
LFcheck 1R	CTTGTCAGTCCCTTCCATTTATTT	
LFcheck 2F	ACACACAGCCAGGGAACGG	1164
LFcheck 2R	GGAGCATTCACTAGGCAACCA	
Fhyg	GTCCTGCGGGTAAATAGCTG	750
Rhyg	ATTTGTGTACGCCCGACAGT	
palF-RTF	GGGTTCCGGGAGAAATGATTAG	102
palF-RTR	GTCTTGCATTCTGTCCGATTTG	

## Data Availability

Data are contained within the article and Appendix A.

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
