# Peer review of "The Arrestin-like Protein palF Contributes to Growth, Sporulation, Spore Germination, Osmolarity, and Pathogenicity of Coniella vitis"

_jof, 2024, doi:10.3390/jof10070508_

Round 1
Reviewer 1 Report
Two major concerns:
1) Authors need to assess the true effect of silencing (not completely) palF expression (expected lowering PalF protein levels) withing the range of phenotypes they describe. Which are due to reducing PacC function (expected from silencing or reducing PalF presence in the cell) and those due to the absence of normal levels of PalF, if these have an specific role. These aspects need to be fully revised.
2) For several results the authors extend their conclusions beyond the reasonable limits of speculations. Also several experiments need of statistical analyses.
1) this study requires of silencing pacC gene in this fungus to understand the role/s of PalF.
2) Results in figure 4 and supplementary figure S3 need of statistical analyses.
3) Authors have to modulate their conclusions in sentences like:
line 237 The wild-type strain showed the capacity to produce acid to decrease the environmental pH, whereas the transformants failed to do so (Figure S3).
No differences are observed and clearing of the dye is observed in the figures for wt and mutant strains.
And this also affects what is shown in discussion:
350 In this study, we found that C. vitis decreased the environmental
351 pH in vitro, and the capability of acidification declined when CvpalF was silenced. These
352 results suggested that CvpalF sensed the environmental pH, and transmits the signal to
353 CvpacC, CvpacC to regulate the expression of genes related to acid production.
Also in discussion
360 Our results suggested that the loss function of PalF affected Pal signal pathway,
361 which regulates ion stress
up to my knowledge there is no clear relationship in other organisms between the absence of ambient pH regulatory system and sensitivity to ion stresses. In fact there are other factors involved. If PalF hypofunctional strain is sensitive to these stresses this needs to be verified by analysing the absence of pacC expression in this fungus.
4) Usually pacC gene is upregulated in response to ambient pH and requires the function of PalF, so it is difficult to understand the expression levels of pacC in figure 7 C. Why pacC is overexpressed in the silencing strains 17 and 29?. Also, FIgure 7A shows an apparent downregulation of the complete set of genes related to ambient pH regulatory system in the absence of palF. Are authors proposing that all these genes are under PacC regulation? or are regulated by other transcription factors? This needs to be verified and this requires the silencing of pacC.
Reviewer 2 Report
If I correctly read the paper jof-3034046-peer-review-v1 “The arrestin-like protein palF plays pivotal roles in growth, development and virulence in Coniella vitis”, the authors report data on the role of arrestin-like protein PalF in Coniella vitis, the aetiological agent of grape white rot. The wild type C. vitis strain GP1 (strain no. 23888, China General Microbiological Culture Collection Center) was used to i) perform bioinformatics analysis on palF gene, ii) construct the CvpalF-silencing plasmid vector, iii) achieve PEG-mediated protoplast transformation, and iv) obtain silenced strains. CvpalF silenced strains were selected on hygromycin (200 μg/mL) and confirmed by PCR and q RT-PCR. The C. vitis strains wild type GP1, the CvpalFi-17 and CvpalFi-29 transformants, and the empty-vector control (CK) were compared for radial growth, sporulation, and spore germination rate under different (3, 5, 7, and 9) pH, in pH shift experiments, and pathogenicity tests performed on detached cv. Red globe grape leaves. Compared to GP1 and CK, CvpalF-silenced strains displayed an acidity-mimicking phenotype, resulting in impaired fungal growth at neutral/alkaline pH, reduced sporulation and pathogenicity, and were hypersensitive to environmental stresses, such as ionic stress, oxidative stress and cell wall stress. CvpalF-silenced strains down-regulated the expression level of genes related to plant cell wall degrading enzymes in an alkaline environment.
Notwithstanding the scientific sound of this work, the presentation in the form of a manuscript lacks scientific strictness. It is difficult to read, confusing, unclear, and requires adjustments.
Verify the correct association number and associated references. E.g., lines 49-50 “Therefore, we speculate that pH is an important factor for the infection of grape by C. vitis [9,13]”, reference “13” is associated with Botrytis cinerea.
Pay attention to punctuation.
In the text, use the acronym after the definition.
The abstract is a very important part of the paper, after reading the manuscript I suggest rewriting it.
Use pertinent keywords.
Line 73: insert a reference after “results”.
Lines 72-80: periods not clear. Rewrite.
Line 96: insert NCBI accession number of PalF protein from other fungal species used in the phylogenetic tree.
Line 110: use “Table”, instead of “table”.
Line 118: What does “YEPD” mean? Define the composition.
Line 119: What does “protoplasm” mean?
Line 122: What does “generations” mean?
Figures and tables must be self-explanatory, clear, and easy to understand without needing any extra explanation. Improve.
Verify the correct association between the Figure number and data presentation.
Line 235: What does “PD” mean?
Line 261-274: How were these experimental data performed?
Figure 1:
Improve the magnification of sections A and B.
Section C:
Insert NCBI Accession number
Use “Neurospora crassa” instead of “Neurosporacrassa”.
Use “Alternaria arborescens” instead of “Alternariaarborescens”.
Improve the legend.
Table 1:
move to the “Materials and Methods” section.
Use “Name” instead of “Primer Name”.
Enlarge the column “Sequence (5’-3’)” and write the palF1 and palF3 sequences in a single row.
If possible, insert information on amplified fragments.
Figure 2:
Improve the magnification of section A
Line 304: What does “transformation” mean?
Improve the legend.
Explain acronyms used: M, 1, 2, 3, 4, 5, 6, CK, CV, WT, …
Figures 3-7:
Improve the legend.
Explain the acronyms used.
If possible, uniform the colours used for the four strains.
Figure 4:
Insert statistical significance.
Figure 5:
Section A: What does “Sporulation (106 spores/mL)” mean?
Figure 6:
Check the correct position of the stressors along the axis.
Improve the discussion section.
Arrange “Reference” following the “Journal of Fungi” guidelines and template (https://www.mdpi.com/journal/jof/instructions).
Lines 379-385: Report this information to the respective figures and tables in Supplementary Materials.
Kind regards
Reviewer 3 Report
The article “The arrestin-like protein palF plays pivotal roles in growth,
development and virulence in Coniella vitis” is devoted to the interesting and acute theme of the interaction of the pathogen Coniella vitis with environmental factors, in particular, pH, and the role of palF in the resistance of the fungus to pH alterations. Authors provide and analyze a lot of information on this theme, but the article makes a doubtful impression since it is carelessly designed.
1)Introduction: Please, insert references on your previous articles (for example, lines 72-73).
2)Materials and methods:
Please, add manufacturer and country in all cases.
2.1. pH of PDA should be added.
2.6. The reference [9] is not accessible, please add another reference and describe the procedure. It is very important, since pH 3 and 9 can delay agar gelation or lead to different solidity of the obtained gels, and, possibly, different accessibility of water and other compounds of medium for the fungus.
2.7. What chamber was used for spores counting?
2.8. Lines 156-157 - Add reference.
3) Results
Fig. 2. I see that transcriptional activity of Cvpa/F gene didn’t change in WT and CK under all pH. Why do authors associate this gene with pH sensitivity? Maybe the housekeeping gene isn’t convenient for this case?
Fig. 3. Transformants show smaller diameter of colonies in all cases of pH, why does the weight of mycelium differ from WT only on mediums with pH 5 and 7?
Fig. 4. I see that authors didn’t find any influences of fungus on pH of the medium. Why are these influences discussed?
Fig. 5. Sporulation of WT and transformants on mediums with pH 5-9 does not differ strongly, authors should check significance of differences.
Fig. 6. The most significant difference between WT and transformants is their contrast ability to grow on the medium with hydrogen peroxide. Maybe the role of CvpalF is reaction to oxidative stress, not alcalinisation?
Discussion: Firstly, authors should clear points presented below and correct doubted arguments. Please, add references on figures after sentences (For example, “We found that C. vitis decreased … (Fig. 2A)).
Introduction: Please, insert references on your previous articles (for example, lines 72-73).
2.8. Lines 156-157 - Add reference.
Author Response
Dear reviewer,
thank you for your careful review and constructive suggestions regarding our manuscript. We have revised the manuscript in accordance with the comments and marked all the amends on our revised manuscript.
Review 3
The article “The arrestin-like protein palF plays pivotal roles in growth,
development and virulence in Coniella vitis” is devoted to the interesting and acute theme of the interaction of the pathogen Coniella vitis with environmental factors, in particular, pH, and the role of palF in the resistance of the fungus to pH alterations. Authors provide and analyze a lot of information on this theme, but the article makes a doubtful impression since it is carelessly designed.
1)Introduction: Please, insert references on your previous articles (for example, lines 72-73).
Answer: Thank you for your suggestion. We had insert references on our previous articles, please see Line 78 in revised manuscript.
2)Materials and methods:
Please, add manufacturer and country in all cases.
Answer: Thank you for your suggestion. We had added manufacturer and country in in revised manuscript.please see Line 155-164.
2.1. pH of PDA should be added.
Answer: Thank you for your suggestion. We had added pH in Line 91 in revised manuscript.
2.6. The reference [9] is not accessible, please add another reference and describe the procedure. It is very important, since pH 3 and 9 can delay agar gelation or lead to different solidity of the obtained gels, and, possibly, different accessibility of water and other compounds of medium for the fungus.
Answer: Here the website to get reference [9]. https://www.sciencedirect.com/science/article/pii/S2095311924000029
And I agree with you that pH 3 and 9 can delay agar gelation or lead to different solidity of the obtained gels, but we inoculated the fungi on the surface of the mediums, so the influence of solidity on the growth of the fungi are slight.
2.7. What chamber was used for spores counting?
Answer: We used hemocytometer for hemocytometer. The large square of the hemocytometer is divided into 16 cells, each cell is divided into 25 smaller cells, that is, 16 × 25 type (Healy format).
2.8. Lines 156-157 - Add reference.
Answer: We had add the reference in line 168 in the revised manuscript.
3) Results
Fig. 2. I see that transcriptional activity of Cvpa/F gene didn’t change in WT and CK under all pH. Why do authors associate this gene with pH sensitivity? Maybe the housekeeping gene isn’t convenient for this case?
Answer: In factly, the exptession of CvpalF gene was different in different pH environment. In this study, relative expression of CvpalF in WT strain in pH 3, 5 ,7, 9 were taken as the value of 1, and the expression of CvpalF in transformants were compared with that in WT in pH 3, 5 ,7, 9, respectively.
Fig. 3. Transformants show smaller diameter of colonies in all cases of pH, why does the weight of mycelium differ from WT only on mediums with pH 5 and 7?
Answer: The growth of Coniella vitis and transformants were both severely restricted in pH 9, so the weight of mycelium differ from WT only on mediums with pH 5 and 7, and slight differ were observed in pH 9.
Fig. 4. I see that authors didn’t find any influences of fungus on pH of the medium. Why are these influences discussed?
Answer: As showed in Figure 4, the pH of culture solution of C. vitis GP1 and transformants had not significantly difference, when they were cultured in PDB (potato dextrose broth) of pH3 and pH9 (Figure 4A,D). the capability of acidification of C. vitis in PD media of pH 5 and pH 7 was slightly declined when CvpalF was silenced (Figure4 B, C).
Fig. 5. Sporulation of WT and transformants on mediums with pH 5-9 does not differ strongly, authors should check significance of differences.
Answer: Thank you for your suggestion. We had revised the significant differences of figure 5A.
Fig. 6. The most significant difference between WT and transformants is their contrast ability to grow on the medium with hydrogen peroxide. Maybe the role of CvpalF is reaction to oxidative stress, not alcalinisation?
Answer: Yes, I agree with you, the most significant difference between WT and transformants is their contrast ability to grow on the medium with hydrogen peroxide, the role of CvpalF is reaction to oxidative stress.
Discussion: Firstly, authors should clear points presented below and correct doubted arguments. Please, add references on figures after sentences (For example, “We found that C. vitis decreased … (Fig. 2A)).
Answer: We had added references on figures after sentence and corrected doubted arguments.
Round 2
Reviewer 1 Report
I note that the manuscript has been modified in some sections, which adds value to this work. The authors argue that deletion/silencing of PacC is dispensable for understanding this work. I have to remind the authors that PalF is a signalling protein and PacC is the effector protein. It is impossible to fully understand the role of PalF without first understanding the role of PacC in this fungus.
Certainly, the description of PalF in C. vitis is of interest and the modification made in the abstract and the changes made in the abstract focus the reader on the aim of this paper.
.
No detailed comments.
Author Response
Thank you for your suggestion. PacC is the effector protein and play important role in regulating growth and virulence in a number of pathogenic fungi. In our present study, we had silenced gene CvpacC also affect the growth and virulence in Coniella vitis, the expression levels of CvpacC increased 2.77 folds under alkaline conditions compared to acidic conditions (unpublished), and over 200 genes related to carboxylic acid metabolism are differentially expressed, in addition, palF affect the expression the pacC, and we had added the discussion of the function palF and pacC in revised manuscript.
Reviewer 2 Report
Notwithstanding the scientific sound of this work, the presentation in the form of a manuscript lacks scientific strictness. It is difficult to read, confusing, unclear, and requires adjustments.
Dear Editor and Authors,
I appreciate the answers to my questions addressed in the cover letter, but were not inserted in the revised manuscript jof-3034046-peer-review-v2.
Many of the suggestions have been included, some have been answered in the cover letter, others remain unanswered.
The revised version of the manuscript remains confusing, unclear, and difficult to read.
Different parts of the manuscript report a eulogy to the authors and their team.
Could you improve the abstract and the introduction sections?
Were cultures on PDB or PDA performed in the dark or under light exposure?
The “Materials and Methods” lacks information on “3.6. The responses of CvpalF to cell wall integrity, oxidation and ionic stress.
Protoplasm (the substances contained in the cell, surrounded by the cell membrane) was prepared on line 127, but protoplasts (fungal or plant cells without the cell wall) were used on lines 129, 214 and 370.
Generations (line 130) should be considered as three successive subcultures starting from mycelia plugs.
The acronym “PD” was not defined at the first mention (line 185). Previously PDA and PDB were used and defined as acronyms and applications. PDA was used at pH = 6.5, for transformants and GP1 cultures, hygromycin B sensitivity assay, growth at different pH values (3, 5, 7, and 9), and sporulation and spore germination rate experiments. PDB was defined at line 92 and re-defined at line 250.
Is PD a new medium?
Why is Table 1 in the “Result” section?
Figures legends remain unclear, unprecise, and difficult to read and understand.
E.g.
Lines 316-316
Figure 1: Bioinformatics analysis of palF genes in different fungal species: protein domain (A), multiple sequence alignment (B), phylogenetic tree (C).
Lines 320-329
Figure 2. Construction and assay of Coniella vitis transformants: (A) Recombinant pSilent-1 vector. (B) PCR profile of Hygromycin (Hyg) resistance gene amplified by primers LFcheck 1F/1R, LFcheck 2F/2R) and Fhyg/Rhyg (Table 1). Lane M: 100-2,000 bp DNA marker, lane 1-6: different C. vitis transformants, CK: C. vitis transformant with empty vector, WT: wild type C. vitis strain GP1. (C) Colony morphology of WT, CK, and C. vitis transformant strains CvpalFi-17 and CvpalFi-29 after 3(?) days on Potato Dextrose Agar at ??°C in the dark(?). (D) Cvpal gene expression in C. vitis WT, CK, CvpalFi-17 and CvpalFi-29 cropped on (?) under different initial pH at (?) °C, for (?) days, in the dark(?). For each pH tested, the CvpalF relative expression in the WT strain was used to normalize the values. Data are the means of three experiments ± se.
How were the results exposed in Figure 2D obtained?
...
Kind regards
Author Response
1. Could you improve the abstract and the introduction sections?
Answer: Thank you for your suggestion. We had improved the abstract and the introduction sections in the revised manuscript.
2. Were cultures on PDB or PDA performed in the dark or under light exposure?
Answer: The fungus was cultured on PDB or PDA in the dark.
3. The “Materials and Methods” lacks information on “3.6. The responses of CvpalF to cell wall integrity, oxidation and ionic stress.
Answer: Thank you for your advice. We had added the method of “Sensitivity of the growth of CvPalF-silenced strains to different osmotic stress” to the section of materials and methods. Please see the line 194-201 in revised manuscript.
4. Protoplasm (the substances contained in the cell, surrounded by the cell membrane) was prepared on line 127, but protoplasts (fungal or plant cells without the cell wall) were used on lines 129, 214 and 370.
Answer: We had change the word “protoplasm” to “protoplasts” in the line 127
5. Generations (line 130) should be considered as three successive subcultures starting from mycelia plugs.
Answer: We had revised the sentence in line 130-131 in the revised manuscript.
6. The acronym “PD” was not defined at the first mention (line 185). Previously PDA and PDB were used and defined as acronyms and applications. PDA was used at pH = 6.5, for transformants and GP1 cultures, hygromycin B sensitivity assay, growth at different pH values (3, 5, 7, and 9), and sporulation and spore germination rate experiments. PDB was defined at line 92 and re-defined at line 250.
Is PD a new medium?
Answer: The PD media should be PDB midia, we had changed “PD” to “PDB” in revised manuscript.
7. Why is Table 1 in the “Result” section?
Answer: We had removed Table 1 to the materials and methods section.
8. Figures legends remain unclear, unprecise, and difficult to read and understand.
E.g.
Lines 316-316
Figure 1: Bioinformatics analysis of palF genes in different fungal species: protein domain (A), multiple sequence alignment (B), phylogenetic tree (C).
Lines 320-329
Figure 2. Construction and assay of Coniella vitis transformants: (A) Recombinant pSilent-1 vector. (B) PCR profile of Hygromycin (Hyg) resistance gene amplified by primers LFcheck 1F/1R, LFcheck 2F/2R) and Fhyg/Rhyg (Table 1). Lane M: 100-2,000 bp DNA marker, lane 1-6: different C. vitis transformants, CK: C. vitis transformant with empty vector, WT: wild type C. vitis strain GP1. (C) Colony morphology of WT, CK, and C. vitis transformant strains CvpalFi-17 and CvpalFi-29 after 3(?) days on Potato Dextrose Agar at ??°C in the dark(?). (D) Cvpal gene expression in C. vitis WT, CK, CvpalFi-17 and CvpalFi-29 cropped on (?) under different initial pH at (?) °C, for (?) days, in the dark(?). For each pH tested, the CvpalF relative expression in the WT strain was used to normalize the values. Data are the means of three experiments ± se.
Answer: We had improved the legend of figure 1-7, and the figure 2 was modified.
9. How were the results exposed in Figure 2D obtained?
Answer: The expression level of the palF-gene in transformants at different environment pH were determined by quantitative real-time PCR (q RT-PCR). C. vitis GP1, CK and CvpalF transformants were cultured in PDB media of pH 3, 5, 7, 9 for 3 days at 28 ℃ with 180 rpm, and the collected the mycelium for RNA extraction and q RT-PCR.To quantify the expression level of the palF-gene in transformants, palF-RTF/R primers (Table S1) were designed for quantitative real-time PCR (q RT-PCR). The actin gene was used as the reference gene, relative expression levels were measured using the 2^(−△△Ct) analysis method.
Reviewer 3 Report
Authors made the great work on the manuscript refinement (matherials and methods details, addition of references, statistical analisys etc), I have received satisfactory answers on all my comments.
In my opinion, the manuscript should be published in present form.
Authors made the great work on the manuscript refinement (matherials and methods details, addition of references, statistical analisys etc), I have received satisfactory answers on all my comments.
In my opinion, the manuscript should be published in present form.
Author Response
Dear reviewers, thank you for your careful review and constructive suggestions regarding our manuscript. As recommended, we have made further revisions to the paper in the light of the reviewer’s comment.
Round 3
Reviewer 2 Report
Dear Editor and Authors,
The revised manuscript jof-3034046-peer-review-v3, remains confused, unclear, dispersive, and difficult to read.
The title is not completely pertinent to the topics of the manuscript.
A possible suggestion:
The Coniella vitis arrestin-like protein palF contributes to growth, sporulation, spore germination and pathogenicity
Line 14: delete “ pH”
Usually, as reported on JoF guidelines at https://www.mdpi.com/journal/jof/instructions, tables and figures are placed in the main text near to the first time they are cited. The editorial staff of the journal will place figures and tables in the appropriate position. In the manuscript, it is difficult to associate the results with the appropriate figures.
The “Materials and Methods” section remains unclear. Experiments carried out are presented in sequence without logical criteria.
As suggested in the previous revisions, the following subdivision would better explain the work done:
2.1. Strains, culture conditions, DNA extraction and bioinformatics analysis
2.2. Construction of the CvpalF-silenced strains
2.2.1. Hygromycin sensitivity
2.2.2. CvpalF-silencing plasmid
2.2.3. Transformation procedure
2.2.4. Gene expression analysis
2.3. Silenced strains characterization
2.3.1. Radial growth under different pH values
2.3.3. Spore production and vitality
2.3.4. Growth in liquid media
2.3.5. Pathogenicity
2.3.6. pH shift experiment
2.3.7. Effects of osmotic stress
2.4. Statistical analysis
The entire “Results” section (lines 213-364) remains unclear and confusing. This section reports unnecessary information on “material and methods” and “discussion”.
E.g.:
lines 218-221 delete “To analyze the homology of PalF proteins, we compared the sequence of CvpalF with the PalF protein sequence in A. nidulans, A. fumigatus, P. oryzae, N. crassa, Verticillium dahliae and Colletotrichum gloeosporioides, and”
lines 212-222: delete “in different fungal was contracted using MEGA 7.0. The results”
Line 221: use “Phylogenetic” instead of “phylogenetic”.
As noted on JoF guidelines at https://www.mdpi.com/journal/jof/instructions, the “Results” section should “provide a concise and precise description of the experimental results, their interpretation, as well as the experimental conclusions that can be drawn”.
Follow the proposed “Material and Method” subdivision to rewrite the “Results” section.
Figure 3-7 legends remain unclear, imprecise, and difficult to read and are not self-explanatory.
Although interesting from a scientific point of view, the present form of the manuscript does not reach the standards for publication on JoF.
Kind regards
See the observations reported in the “major comments” section.
Author Response
Dear reviewer,
Thank you for reviewing our manuscript and for the constructive comments, which greatly helped us to improve the manuscript.
The changes within the revised manuscript are summarized in “Author Response”. We answered the comments point-by-point.
Author Response
- The title is not completely pertinent to the topics of the manuscript.
A possible suggestion:
The Coniella vitis arrestin-like protein palF contributes to growth, sporulation, spore germination and pathogenicity
Answer: Thank you for your suggestion, we had revised the title as recommended.
- Line 14: delete “ pH”
Answer: Thank you for your suggestion, we had deleted “pH” in the line 14 in revised manuscript.
- Usually, as reported on JoF guidelines at https://www.mdpi.com/journal/jof/instructions, tables and figures are placed in the main text near to the first time they are cited. The editorial staff of the journal will place figures and tables in the appropriate position. In the manuscript, it is difficult to associate the results with the appropriate figures.
Answer: Thank you for your advice, we had placed the figures and tables in the appropriate position in the revised manuscript.
- The “Materials and Methods” section remains unclear. Experiments carried out are presented in sequence without logical criteria.
As suggested in the previous revisions, the following subdivision would better explain the work done:
2.1. Strains, culture conditions, DNA extraction and bioinformatics analysis
2.2. Construction of the CvpalF-silenced strains
2.2.1. Hygromycin sensitivity
2.2.2. CvpalF-silencing plasmid
2.2.3. Transformation procedure
2.2.4. Gene expression analysis
2.3. Silenced strains characterization
2.3.1. Radial growth under different pH values
2.3.3. Spore production and vitality
2.3.4. Growth in liquid media
2.3.5. Pathogenicity
2.3.6. pH shift experiment
2.3.7. Effects of osmotic stress
2.4. Statistical analysis
Answer: Thank you for your suggestion, the material and methods section changed as suggested.
- The entire “Results” section (lines 213-364) remains unclear and confusing. This section reports unnecessary information on “material and methods” and “discussion”.
E.g.:
lines 218-221 delete “To analyze the homology of PalF proteins, we compared the sequence of CvpalF with the PalF protein sequence in A. nidulans, A. fumigatus, P. oryzae, N. crassa, Verticillium dahliae and Colletotrichum gloeosporioides, and”
Answer: Thank you for your suggestion, We had deleted the sentence in the line 209 in revised manuscript.
lines 212-222: delete “in different fungal was contracted using MEGA 7.0. The results”
Answer: Thank you for your suggestion, We had deleted the sentence in the line 220 in revised manuscript.
Line 221: use “Phylogenetic” instead of “phylogenetic”.
Answer: Thank you for your suggestion, We had changed “phylogenetic” to “Phylogenetic” in the line 219 in revised manuscript.
6. As noted on JoF guidelines at https://www.mdpi.com/journal/jof/instructions, the “Results” section should “provide a concise and precise description of the experimental results, their interpretation, as well as the experimental conclusions that can be drawn”.
Answer: Thank you for your suggestion, we had added the conclusions section in revised manuscript.
7. Follow the proposed “Material and Method” subdivision to rewrite the “Results” section.
Answer: Thank you for your advice, we had revised the results section according to your suggestions.
8. Figure 3-7 legends remain unclear, imprecise, and difficult to read and are not self-explanatory.
Answer:We had improved the legend of figure 3-7.
Round 4
Reviewer 2 Report
Dear Editor and Authors,
The revised manuscript jof-3034046-peer-review-v4 remains confused, unclear, dispersive, and difficult to read.
Some suggestions to integrate the manuscript and give the possibility of publication on JoF.
Line 2: add “, osmolarity” between “germination” and “and”.
Line 19: use “the wild type (WT) and the empty vector control (CK) strains” instead of “the wild … strains”
Lines 19-20: What the sentence means?
The manuscript lacks information on “The distance between the hyphal branches was significantly increased in the CvpalF-silenced strains.”
How was the distance calculated? In which experiment?
Line 32: delete “all”.
Line 33: use “20-30%” instead of “20%-30%”.
Line 34: use “30-50%” instead of “30%-50%”.
Line 43: use “fungal plant pathogen.” instead of “phytopathogen.”.
Line 49: delete “Studies have shown that”.
Lines 77-86: Periods are not clear. Rewrite.
A possible suggestion:
Previous research identified in C. vitis CvpalF a homologue of PalF/Rim8, with a down-regulated expression at alkaline pH [9]. The present study (i) compares the sequence identity and genetic relationship between CvpalF and other pathogenic fungi; (ii) silences the CvpalF gene via RNA interference; (iii) elucidates the effects of CvpalF on growth, development and stress response under different pH conditions and pathogenicity; (iv) demonstrates the regulation of CvpalF on genes in Pal signaling pathway and other C. vitis virulence genes. Findings provide new insights into the pathogenic mechanism of C. vitis regulated by pH.
Material and Methods section
This section lacks information on:
Figure S3: Medium acidification.
Figure S4: Reduce rate of lesion diameters on grape leaves.
Figure S5: Inhibition rate under different environmental stresses.
Lines 92-93: Were cultures on PDB or PDA performed in the dark or under light exposure? Evidently, it is in the dark, as reported in your previous response to the reviewer.
Were the PCR conditions reported on lines 149-150 used for all PCR and qRT-PCR in the text? Please associate the used condition on all performed amplifications.
Improve information and primers on qRT-PCR experiments reported on lines 198-207
Lines 91-97: Rewrite.
A possible suggestion:
The C. vitis wild-type strain GP1 and transformants generated in this study were cultured on potato dextrose agar (PDA, Difco, Detroit; pH = 6.5) plates at 28 ℃ in the dark for 3 days and potato dextrose broth (PDB) at 28 ℃, in the dark, and 180 rpm.
For genomic DNA extraction, the fungal strains were grown on PDA (28 ℃, darkness) for 3 days on the surface of cellophane membranes inoculated with 4-mm mycelial plugs. Genomic DNA was extracted with the DNeasy Plant Mini Kit (Qiagen) according to the manufacturer’s instructions.
Lines 109-113: Rewrite.
A possible suggestion:
C. vitis GP1 was cultured on PDA for 3 days at 28 ℃ in the dark, following this, mycelial plugs (5 mm diameter) were transferred to fresh PDA containing hygromycin B (Roche) at 0, 1, 5, 10, 50, 100, 200, and 300 μg/mL. The plates were incubated at 28 ℃ in the dark for 3 d. The diameters of developed colonies were measured.
Line 116: use “amplified using the primers” instead of “amplificated used the primers of”.
Insert amplification conditions.
Line 124: insert “(Table 1)” between “2F/R” and “and”.
Line 130: Were cultures on YEPD performed in the dark or under light exposure?
Lines 131-137. Rewrite.
A possible suggestion:
The silencing vector pCvpalFi was transferred into the C. vitis GP1 strain by PEG-mediated protoplast transformation [26,27]. Transformants were selected and cultured on PDA supplemented with hygromycin B (200 μg/mL) for three successive subcultures. After extraction using EZ-DNAaway RNA Mini-Preps Kit (Sangon Biotech, Shanghai, China) according to the manufacturing protocol, CvpalF silenced strains were confirmed by PCR amplification with LFcheck 1F/R, LFcheck 2F/R, Fhyg/Rhyg primers (Table 1).
Lines 139-144. Rewrite.
A possible suggestion:
C. vitis GP1 wild type (WT), GP1 empty vector control (CK) and two CvpalF transformants, CvpalFi-17 and CvpalFi-29, were cultured in PDB at pH 3, 5, 7, 9 for 3 days at 28 ℃, in the dark, 180 rpm. The pH was adjusted using 0.2 M Na2HPO4 12H2O, 0.1 M C6H8O·7H2O, and NaOH as described by Yuan et al. [9]. Total RNA was extracted using EZ-10 DNAaway RNA Mini-Preps Kit according to the manufacturing protocol. RNA samples (1 μg) were reverse transcribed using the HiScript III 1st Strand cDNA Synthesis Kit (+gDNA wiper, Vazyme, Nanjing, China).
Line 156: insert “in the dark” between “°C” and “for”
Line 154-208. Rewrite.
Line 164: Indicate the initial pH of PDB.
Line 166: How were mycelia collected?
Line 185: Indicate the dark/light alternation in the pathogenicity test.
Lines 190-196: Is a control necessary for the experiment?
A possible suggestion:
2.3.1. Growth under different pH
2.3.1.a. On solid media
Strains WT, CK, and CvpalF transformants CvpalFi-17 and CvpalFi-29 were cultured on PDA amended to pH 3, 5, 7, and 9. The pH was adjusted as described by Yuan et al. [9]. After 3 days of incubation (28 ℃, in the dark) the colony diameters were measured, and the colony edges were observed by a light microscope (TESCAN VEGA3 SBU). The experiment was performed with three biological replicates and repeated three times.
2.3.1.b. In liquid media
Spore suspension of C. vitis strains WT, CK, and CvpalF transformants (CvpalFi-17 and CvpalFi-29) was prepared as described previously. PDB (100 mL; pH ???) inoculated with 1mL spore suspension (1 × 106 spore/mL) were cultured at 28 ℃, darkness, at 180 rpm. After 3 days, the mycelia mat was harvested by ????, dried at 60 ℃, and weighed with an analytical balance to determine the mycelia biomass. The pH of each culture was measured every day with a pH meter (Sartorius, PB-10).
2.3.2. Spore production and vitality
C. vitis strains WT, CK, and CvpalF transformants (CvpalFi-17 and CvpalFi-29) were cultured (28 ℃, darkness) on PDA at pH 3, 5, 7, and 9 for 14 days. Spores were collected by washing each colony surface with 40 mL of 0.05% (v/v) Tween 80 (Sigma‒Aldrich, Copenhagen, Denmark) in distilled water. Suspensions were filtered on miracloth (Calbiochem) into 50 mL tubes. The spore concentrations were determined by hemocytometer, and spore morphology was observed under light microscopy (Olympus).
Spore vitality was verified as proposed by Yuan et al. [9].
EXPLAIN SPORE PRODUCTION, COLLECTION AND VITALIY ASSAY ACCORDING TO FIGURE 4B.
2.3.3. Pathogenicity
Mycelia plugs (5 mm diameter) taken from 3-days-old C. vitis strains WT, CK, and CvpalF transformants (CvpalFi-17 and CvpalFi-29) colonies on PDA incubated (28 ℃, darkness) were used to inoculate detached grape leaves (cv. Red globe). Detached grape leaves inoculated with PDA plugs were performed as a control. Inoculated leaves were incubated in a greenhouse at 28 ℃, 90% RH and ??? natural photoperiod. The spot diameters on the leaves were determined at 1, 2, and 3 dpi (day post inoculation). The experiment was repeated three times, and three biological replicates were performed.
2.3.4. Effects of osmotic stress
The effects of CvpalF on osmotic stress adaptability were performed on PDA amended with KCl (1 M), NaCl (1 M), CaSO4 (1 M), MgSO4 (1 M), H2O2 (1 M) or Congo red (1 M). Plates were inoculated with a mycelial plug (5 mm diameter) cut from 3-days-old C. vitis strains WT, CK, CvpalFi-17 and CvpalFi-29 cultures on PDA (28 ℃, in the dark). After 7 days of incubation (28 ℃, in the dark) the colony diameter was measured.
2.3.5. Virulence gene expression
Conidia of C. vitis strains WT, CK, CvpalFi-17 and CvpalFi-29 were inoculated in 20 ml of PDB with a pH of 3.0 to obtain a final concentration of 3 × 106 spores/mL. Flasks were incubated at 28 ℃, in the dark, with shaking at 180 rpm for 24 h. Then the cultures were shifted to PDB with pH values of 3.0 and pH 9.0 for further 15, 30, 60, or 120 min according to the method of Hervas-Aguilar et al. [29]. Subsequently, mycelia were harvested using four layers of cheesecloth. Total RNA was extracted, and qRT-PCR was carried out as described by Yuan et al. [9] and Mushtaq et al. [30] using the primer pairs of the Pal signaling pathway and other virulence genes listed in Table S1.
A DESCRIPTION OF THE GENE USED IS SUITABLE. SEE FIGURE 8.
2.3. Statistical analysis
Results
Line 220: add “(Figure 1C)” between “proteins” and “showed”.
Line 222: delete “ (Figure 1C)”.
Lines 230-244. Rewrite.
A possible suggestion:
The pCvpalFi vector and the pSlient-1 vector used as a vacant vector control (Figure 2A, Figure S1) were inserted via PEG-mediated protoplast transformation into WT C. vitis strain GP1.
Hygromycin B fully suppressed the growth of C. vitis GP1 at 100 μg/mL (Figure S2). To avoid false positives, Hygromycin B at 200 μ g/mL was used for selecting transformants. Six transformants were randomly selected. PCR amplifications with primers LFcheck 1F/1R, LFcheck 2F/2R, and F/Rhyg confirm transformation, yielding bands of 1072, 1164, and 750 bp, respectively (Figure 2B).
The expression level of the CvpalF gene was significantly reduced in the CvpalFi-17 and CvpalFi-29 transformants, with gene silencing efficiency of 76.17 – 98.93 % (Figure 2C,D). Meanwhile, the expression level of the CvpalF gene was not significantly changed in CK and WT strains.
Figure 2C:
What “CV” mean? Replace with WT or explain it in the legend.
Lines 269-283. Rewrite.
A possible suggestion:
3.3.1. Growth under different pH values
CvpalF silenced strains CvpalFi-17 and CvpalFi-29 exhibited impaired growth compared with WT and CK with significant inhibition under alkaline conditions (Figure 3A). Compared to WT and CK strains, the growth inhibition (Figure 3B) of the two CvpalF silenced strains ranged from 28.48 to 46.81%. Additionally, the colony edges appeared shorter and sparser in CvpalFi-17 and CvpalFi-29 (Figure 3C).
The mycelial biomass of strains CvpalFi-17 and CvpalFi-29 was significantly reduced by the pH of PDB (Figure 3D). The mycelial biomasses of CvpalF silenced strains CvpalFi-17 and CvpalFi-29 declined 20.86 and 14.75%, 65.71 and 66.94%, 52.06 and 58.22% at pH 3, 5, and 7, respectively (Figure 3D). However, the mycelial weight of CvpalFi-17 and CvpalFi-29 had no significant reduction when grown in PDB at pH 9 (Figure 3D).
Lines 286-293: Rewrite.
A possible suggestion:
Figure 3.
Coniella vitis strain GP1 wild type (WT), GP1 empty vector transformant control (CK) and two CvpalF silenced strains (CvpalFi-17 and CvpalFi-29) growth on Potato Dextrose Agar (PDA) or in Potato Dextrose Broth (PDB) under pH 3, 5,7 or 9: colony appearance (A), diameter (B) and hyphae morphology (C) on PDA, mycelial biomass on PDB (D). For culture conditions see the material and methods section. Histograms report the average of nine replicates ± SE. In each chart, different letters indicate significant differences (P < 0.05) according to Tukey’s test.
Lines 294-303: Improve.
Figure 4: insert “Incubation times (d)” under the abscissa axis.
Lines 305-308: Rewrite.
A possible suggestion:
Figure 4. Variation of pH in Potato Dextrose Broth (PDB) during the growth of Coniella vitis strain GP1 (WT), GP1 empty vector transformant (CK) and two CvpalF silenced strains (CvpalFi-17 and CvpalFi-29). pH3, pH5, pH7 and pH9 indicate the initial pH of PDB. Values are the average of nine replicates ± SE. In each chart, different letters indicate significant differences according to Tukey’s test at P < 0.05.
Lines 309-321. Improve.
Figure 5:
Increase the size of the “sporulation” axis in section “A”.
Is it possible to write the labels of the abscissa horizontally?
Lines 323-327: Rewrite.
A possible suggestion:
Figure 5. Coniella vitis strain GP1 (WT), GP1 empty vector transformant (CK) and two CvpalF silenced strains (CvpalFi-17 and CvpalFi-29) spore production (A) from 14-day-old culture on Potato Dextrose Agar (28 °C, in the dark). Spore germination rate after 16 h of incubation (28 °C, in the dark) in 10 % grape juice. Values are the average of nine replicates ± SE. In each chart, different letters indicate significant differences according to Tukey’s test at P < 0.05.
Lines 328-336: Improve.
A possible suggestion:
3.3.3. Pathogenicity test
On detached leaves of cv. Red globe, CvpalF silenced strains CvpalFi-17 and CvpalFi-29, developed brown spots much slower than strains WT and CK. Compared with WT and CK, a reduction of spot diameter by 55.75 and 32.87 % in strains CvpalFi-17 and CvpalFi-29, respectively, was recorded (Figure 6A,B and Figure S4).
Lines 337-341: Improve.
A possible suggestion:
Figure 6. Pathogenicity test of Coniella vitis strain GP1 (WT), GP1 empty vector transformant (CK) and two CvpalF silenced strains (CvpalFi-17 and CvpalFi-29) on Vitis vinifera cv Red globe on detached leaves: (A) symptoms and lesion diameters (B) development. Histograms report the average of nine replicates ± SE, different letters indicate significant differences (P < 0.05) according to Tukey’s test.
Lines 342-354: Improve.
Lines 356-360: Improve.
A possible suggestion:
Figure 7. Response to different stresses of Coniella vitis strain GP1 (WT), GP1 empty vector transformant (CK) and two CvpalF silenced strains (CvpalFi-17 and CvpalFi-29) growth for 7 days (28 ℃, in the dark) on Potato Dextrose Agar amended with 1M KCl, NaCl, CaSO4, MgSO4, H2O2 or Congo red: (A) Colony morphologies and (B) Colony diameter. Histograms report the average of nine replicates ± SE, different letters indicate significant differences (P < 0.05) according to Tukey’s test.
Lines 361-380: Improve.
Lines 382-385: Improve.
Explain the significance of sections A, B and C.
A possible suggestion:
Figure 8. Coniella vitis strain GP1 (WT), and two CvpalF silenced strains (CvpalFi-17 and CvpalFi-29) gene expression: A) genes related to Pal signaling pathway, B) plant cell wall degrading enzymes, and C) ????? . Relative expression was normalized using actin as an internal control. The relative gene expression levels in WT were taken as the value of 1. The primers used are listed in Table S1.
Discussion
Lines 394-399: use “In this study, a homologous gene of PalF/Rim8 was identified in C. vitis. The CvpalF silenced strains CvpalFi-17 and CvpalFi-29 were also obtained using an RNAi approach with the fungal transformation vector pSilent-1 through PEG-mediated protoplast transformation. The applied method was successful in both Ascomycota and Basidiomycota [37–39]. Furthermore, the role of CvpalF in growth, virulence, and response to the different pH conditions was elucidated. Compared with C. vitis WT and CK, CvpalF silenced strains reduce linear growth, mycelia biomass production and worse pathogenicity.” instead of “In this study … Figure 5C).”.
Line 410: use “Previous” instead of “Our previous”.
Lines 413-422: Improve.
Lines 428-436: Improve.
Line 343: delete (ROS). The abbreviation was not used.
Lines 460-472: Improve.
Line 484: use “contained” instead of “c.ontained”.
Kind regards
See Detail comments section.
Author Response

(The authors gave the same response as above.)

Author Response
Line 2: add “, osmolarity” between “germination” and “and”.
Answer: Answer: Thank you for your suggestion, we had added “osmolarity” in the title.
Line 19: use “the wild type (WT) and the empty vector control (CK) strains” instead of “the wild … strains”
Answer: We had revised the sentence as suggested.
Lines 19-20: What the sentence means?
The manuscript lacks information on “The distance between the hyphal branches was significantly increased in the CvpalF-silenced strains.”
How was the distance calculated? In which experiment?
Answer: We had added the sentence in the line 20-21 in revised manuscript.
Line 32: delete “all”.
Answer: We had deleted “all” in the line 33 in revised manuscript.
Line 33: use “20-30%” instead of “20%-30%”.
Line 34: use “30-50%” instead of “30%-50%”.
Answer: We had used “20-30%” and “30-50%” instead of “20%-30%” and “30%-50%” in the line 34 in revised manuscript.
Line 43: use “fungal plant pathogen.” instead of “phytopathogen.”.
Answer: We had used “fungal plant pathogen.” instead of “phytopathogen.” in the line 43 in revised manuscript.
Line 49: delete “Studies have shown that”.
Answer: We had deteled “Studies have shown that” in the line 50 in revised manuscript.
Lines 77-86: Periods are not clear. Rewrite.
A possible suggestion:
Previous research identified in C. vitis CvpalF a homologue of PalF/Rim8, with a down-regulated expression at alkaline pH [9]. The present study (i) compares the sequence identity and genetic relationship between CvpalF and other pathogenic fungi; (ii) silences the CvpalF gene via RNA interference; (iii) elucidates the effects of CvpalF on growth, development and stress response under different pH conditions and pathogenicity; (iv) demonstrates the regulation of CvpalF on genes in Pal signaling pathway and other C. vitis virulence genes. Findings provide new insights into the pathogenic mechanism of C. vitis regulated by pH.
Answer: Answer: Thank you for your suggestion, we had revised the paragraph in the line 77-84 in revised manuscript.
Material and Methods section
This section lacks information on:
Figure S3: Medium acidification.
Figure S4: Reduce rate of lesion diameters on grape leaves.
Figure S5: Inhibition rate under different environmental stresses.
Answer: The lacks information is in the Supplementary Materials section.
Lines 92-93: Were cultures on PDB or PDA performed in the dark or under light exposure? Evidently, it is in the dark, as reported in your previous response to the reviewer.
Answer: The strains were cultured in the dark. We had revised the sentence in the line 90-91 in revised manuscript.
Were the PCR conditions reported on lines 149-150 used for all PCR and qRT-PCR in the text? Please associate the used condition on all performed amplifications.
Answer: We We had revised the paragraph in the line 136-138 in revised manuscript.
Improve information and primers on qRT-PCR experiments reported on lines 198-207
Answer: We We had revised the paragraph in the line 202-211 in revised manuscript.
Lines 91-97: Rewrite.
A possible suggestion:
The C. vitis wild-type strain GP1 and transformants generated in this study were cultured on potato dextrose agar (PDA, Difco, Detroit; pH = 6.5) plates at 28 ℃ in the dark for 3 days and potato dextrose broth (PDB) at 28 ℃, in the dark, and 180 rpm.
For genomic DNA extraction, the fungal strains were grown on PDA (28 ℃, darkness) for 3 days on the surface of cellophane membranes inoculated with 4-mm mycelial plugs. Genomic DNA was extracted with the DNeasy Plant Mini Kit (Qiagen) according to the manufacturer’s instructions.
Answer: We had revised the sentence as suggested. Please see the line 89-96 in revised manuscript.
Lines 109-113: Rewrite.
A possible suggestion:
- vitis GP1 was cultured on PDA for 3 days at 28 ℃ in the dark, following this, mycelial plugs (5 mm diameter) were transferred to fresh PDA containing hygromycin B (Roche) at 0, 1, 5, 10, 50, 100, 200, and 300 μg/mL. The plates were incubated at 28 ℃ in the dark for 3 d. The diameters of developed colonies were measured.
Answer: We had revised the sentence as suggested. Please see the line 108-112 in revised manuscript.
Line 116: use “amplified using the primers” instead of “amplificated used the primers of”.
Answer: We had used “amplified using the primers” instead of “amplificated used the primers of in the line 115 in revised manuscript.
Insert amplification conditions.
Answer: We had added the amplification conditions in the line 136-138 in revised manuscript.
Line 124: insert “(Table 1)” between “2F/R” and “and”.
Answer: We had inserted “Table 1” in the line 123 in revised manuscript.
Line 130: Were cultures on YEPD performed in the dark or under light exposure?
Answer: The strains were cultured in the dark. We had revised the sentence in the line 128 in revised manuscript.
Lines 131-137. Rewrite.
A possible suggestion:
The silencing vector pCvpalFi was transferred into the C. vitis GP1 strain by PEG-mediated protoplast transformation [26,27]. Transformants were selected and cultured on PDA supplemented with hygromycin B (200 μg/mL) for three successive subcultures. After extraction using EZ-DNAaway RNA Mini-Preps Kit (Sangon Biotech, Shanghai, China) according to the manufacturing protocol, CvpalF silenced strains were confirmed by PCR amplification with LFcheck 1F/R, LFcheck 2F/R, Fhyg/Rhyg primers (Table 1).
Answer: Thank you for your suggestion. We had revised the paragraph as suggested. Please see the line 132-139 in revised manuscript.
Lines 139-144. Rewrite.
A possible suggestion:
- vitis GP1 wild type (WT), GP1 empty vector control (CK) and two CvpalF transformants, CvpalFi-17 and CvpalFi-29, were cultured in PDB at pH 3, 5, 7, 9 for 3 days at 28 ℃, in the dark, 180 rpm. The pH was adjusted using 0.2 M Na2HPO4 12H2O, 0.1 M C6H8O·7H2O, and NaOH as described by Yuan et al. [9]. Total RNA was extracted using EZ-10 DNAaway RNA Mini-Preps Kit according to the manufacturing protocol. RNA samples (1 μg) were reverse transcribed using the HiScript III 1st Strand cDNA Synthesis Kit (+gDNA wiper, Vazyme, Nanjing, China).
Answer: Thank you for your suggestion. We had revised the paragraph as suggested. Please see the line 140-143 in revised manuscript.
Line 156: insert “in the dark” between “°C” and “for”
Answer: We had inserted “in the dark” in the line 142 in revised manuscript.
Line 154-208. Rewrite.
Answer: We had revised the paragraph as suggested. Please see the line 157-211 in revised manuscript.
Line 164: Indicate the initial pH of PDB.
Answer: We had added the initial pH of PDB in the line 171 in revised manuscript.
Line 166: How were mycelia collected?
Answer: We added the method of mycelia collected in the line 172-175 in revised manuscript.
Line 185: Indicate the dark/light alternation in the pathogenicity test.
Answer: We had revised the sentence in the line 189 in revised manuscript.
Lines 190-196: Is a control necessary for the experiment?
Answer: The control is not necessary for the experiment.
A possible suggestion:
2.3.1. Growth under different pH
2.3.1.a. On solid media
Strains WT, CK, and CvpalF transformants CvpalFi-17 and CvpalFi-29 were cultured on PDA amended to pH 3, 5, 7, and 9. The pH was adjusted as described by Yuan et al. [9]. After 3 days of incubation (28 ℃, in the dark) the colony diameters were measured, and the colony edges were observed by a light microscope (TESCAN VEGA3 SBU). The experiment was performed with three biological replicates and repeated three times.
2.3.1.b. In liquid media
Spore suspension of C. vitis strains WT, CK, and CvpalF transformants (CvpalFi-17 and CvpalFi-29) was prepared as described previously. PDB (100 mL; pH ???) inoculated with 1mL spore suspension (1 × 106 spore/mL) were cultured at 28 ℃, darkness, at 180 rpm. After 3 days, the mycelia mat was harvested by ????, dried at 60 ℃, and weighed with an analytical balance to determine the mycelia biomass. The pH of each culture was measured every day with a pH meter (Sartorius, PB-10).
2.3.2. Spore production and vitality
- vitis strains WT, CK, and CvpalF transformants (CvpalFi-17 and CvpalFi-29) were cultured (28 ℃, darkness) on PDA at pH 3, 5, 7, and 9 for 14 days. Spores were collected by washing each colony surface with 40 mL of 0.05% (v/v) Tween 80 (Sigma‒Aldrich, Copenhagen, Denmark) in distilled water. Suspensions were filtered on miracloth (Calbiochem) into 50 mL tubes. The spore concentrations were determined by hemocytometer, and spore morphology was observed under light microscopy (Olympus).
Spore vitality was verified as proposed by Yuan et al. [9].
EXPLAIN SPORE PRODUCTION, COLLECTION AND VITALIY ASSAY ACCORDING TO FIGURE 4B.
2.3.3. Pathogenicity
Mycelia plugs (5 mm diameter) taken from 3-days-old C. vitis strains WT, CK, and CvpalF transformants (CvpalFi-17 and CvpalFi-29) colonies on PDA incubated (28 ℃, darkness) were used to inoculate detached grape leaves (cv. Red globe). Detached grape leaves inoculated with PDA plugs were performed as a control. Inoculated leaves were incubated in a greenhouse at 28 ℃, 90% RH and ??? natural photoperiod. The spot diameters on the leaves were determined at 1, 2, and 3 dpi (day post inoculation). The experiment was repeated three times, and three biological replicates were performed.
2.3.4. Effects of osmotic stress
The effects of CvpalF on osmotic stress adaptability were performed on PDA amended with KCl (1 M), NaCl (1 M), CaSO4 (1 M), MgSO4 (1 M), H2O2 (1 M) or Congo red (1 M). Plates were inoculated with a mycelial plug (5 mm diameter) cut from 3-days-old C. vitis strains WT, CK, CvpalFi-17 and CvpalFi-29 cultures on PDA (28 ℃, in the dark). After 7 days of incubation (28 ℃, in the dark) the colony diameter was measured.
2.3.5. Virulence gene expression
Conidia of C. vitis strains WT, CK, CvpalFi-17 and CvpalFi-29 were inoculated in 20 ml of PDB with a pH of 3.0 to obtain a final concentration of 3 × 106 spores/mL. Flasks were incubated at 28 ℃, in the dark, with shaking at 180 rpm for 24 h. Then the cultures were shifted to PDB with pH values of 3.0 and pH 9.0 for further 15, 30, 60, or 120 min according to the method of Hervas-Aguilar et al. [29]. Subsequently, mycelia were harvested using four layers of cheesecloth. Total RNA was extracted, and qRT-PCR was carried out as described by Yuan et al. [9] and Mushtaq et al. [30] using the primer pairs of the Pal signaling pathway and other virulence genes listed in Table S1.
A DESCRIPTION OF THE GENE USED IS SUITABLE. SEE FIGURE 8.
2.3. Statistical analysis
Results
Line 220: add “(Figure 1C)” between “proteins” and “showed”.
Answer: We had added “(Figure 1C)” in the line 224 in revised manuscript.
Line 222: delete “ (Figure 1C)”.
Answer: We had deleted “(Figure 1C)” in the line 226 in revised manuscript.
Lines 230-244. Rewrite.
A possible suggestion:
The pCvpalFi vector and the pSlient-1 vector used as a vacant vector control (Figure 2A, Figure S1) were inserted via PEG-mediated protoplast transformation into WT C. vitis strain GP1.
Hygromycin B fully suppressed the growth of C. vitis GP1 at 100 μg/mL (Figure S2). To avoid false positives, Hygromycin B at 200 μ g/mL was used for selecting transformants. Six transformants were randomly selected. PCR amplifications with primers LFcheck 1F/1R, LFcheck 2F/2R, and F/Rhyg confirm transformation, yielding bands of 1072, 1164, and 750 bp, respectively (Figure 2B).
The expression level of the CvpalF gene was significantly reduced in the CvpalFi-17 and CvpalFi-29 transformants, with gene silencing efficiency of 76.17 – 98.93 % (Figure 2C,D). Meanwhile, the expression level of the CvpalF gene was not significantly changed in CK and WT strains.
Answer: Thank you for your suggestion. We had revised the paragraph as suggested. Please see the line 230-240 in revised manuscript.
Figure 2C:
What “CV” mean? Replace with WT or explain it in the legend.
Answer: I am sorry for our mistake, we had changed “CV” to “WT” in Figure 2C.
Lines 269-283. Rewrite.
A possible suggestion:
3.3.1. Growth under different pH values
CvpalF silenced strains CvpalFi-17 and CvpalFi-29 exhibited impaired growth compared with WT and CK with significant inhibition under alkaline conditions (Figure 3A). Compared to WT and CK strains, the growth inhibition (Figure 3B) of the two CvpalF silenced strains ranged from 28.48 to 46.81%. Additionally, the colony edges appeared shorter and sparser in CvpalFi-17 and CvpalFi-29 (Figure 3C).
The mycelial biomass of strains CvpalFi-17 and CvpalFi-29 was significantly reduced by the pH of PDB (Figure 3D). The mycelial biomasses of CvpalF silenced strains CvpalFi-17 and CvpalFi-29 declined 20.86 and 14.75%, 65.71 and 66.94%, 52.06 and 58.22% at pH 3, 5, and 7, respectively (Figure 3D). However, the mycelial weight of CvpalFi-17 and CvpalFi-29 had no significant reduction when grown in PDB at pH 9 (Figure 3D).
Answer: Thank you for your suggestion. We had revised the paragraph as suggested. Please see the line 252-260 in revised manuscript.
Lines 286-293: Rewrite.
A possible suggestion:
Figure 3.
Coniella vitis strain GP1 wild type (WT), GP1 empty vector transformant control (CK) and two CvpalF silenced strains (CvpalFi-17 and CvpalFi-29) growth on Potato Dextrose Agar (PDA) or in Potato Dextrose Broth (PDB) under pH 3, 5,7 or 9: colony appearance (A), diameter (B) and hyphae morphology (C) on PDA, mycelial biomass on PDB (D). For culture conditions see the material and methods section. Histograms report the average of nine replicates ± SE. In each chart, different letters indicate significant differences (P < 0.05) according to Tukey’s test.
Answer: Thank you for your suggestion. We had revised the paragraph as suggested. Please see the line 264-269 in revised manuscript.
Lines 294-303: Improve.
Answer: We had revised the paragraph in the line 271-278 in revised manuscript.
Figure 4: insert “Incubation times (d)” under the abscissa axis.
Answer: We had add the “Incubation times (d)” under the abscissa axis in Figure 4
Lines 305-308: Rewrite.
A possible suggestion:
Figure 4. Variation of pH in Potato Dextrose Broth (PDB) during the growth of Coniella vitis strain GP1 (WT), GP1 empty vector transformant (CK) and two CvpalF silenced strains (CvpalFi-17 and CvpalFi-29). pH3, pH5, pH7 and pH9 indicate the initial pH of PDB. Values are the average of nine replicates ± SE. In each chart, different letters indicate significant differences according to Tukey’s test at P < 0.05.
Answer: Thank you for your suggestion. We had revised the paragraph as suggested. Please see the line 280-284 in revised manuscript.
Lines 309-321. Improve.
Answer: We had revised the paragraph in the line 287-297 in revised manuscript.
Figure 5:
Increase the size of the “sporulation” axis in section “A”.
Is it possible to write the labels of the abscissa horizontally?
Answer: We had revised the axis of Figure 5.
Lines 323-327: Rewrite.
A possible suggestion:
Figure 5. Coniella vitis strain GP1 (WT), GP1 empty vector transformant (CK) and two CvpalF silenced strains (CvpalFi-17 and CvpalFi-29) spore production (A) from 14-day-old culture on Potato Dextrose Agar (28 °C, in the dark). Spore germination rate after 16 h of incubation (28 °C, in the dark) in 10 % grape juice. Values are the average of nine replicates ± SE. In each chart, different letters indicate significant differences according to Tukey’s test at P < 0.05.
Answer: We had revised the paragraph as suggested. Please see the line 299-304 in revised manuscript.
Lines 328-336: Improve.
A possible suggestion:
3.3.3. Pathogenicity test
On detached leaves of cv. Red globe, CvpalF silenced strains CvpalFi-17 and CvpalFi-29, developed brown spots much slower than strains WT and CK. Compared with WT and CK, a reduction of spot diameter by 55.75 and 32.87 % in strains CvpalFi-17 and CvpalFi-29, respectively, was recorded (Figure 6A,B and Figure S4).
Answer: We had revised the paragraph as suggested. Please see the line 305-309 in revised manuscript.
Lines 337-341: Improve.
A possible suggestion:
Figure 6. Pathogenicity test of Coniella vitis strain GP1 (WT), GP1 empty vector transformant (CK) and two CvpalF silenced strains (CvpalFi-17 and CvpalFi-29) on Vitis vinifera cv Red globe on detached leaves: (A) symptoms and lesion diameters (B) development. Histograms report the average of nine replicates ± SE, different letters indicate significant differences (P < 0.05) according to Tukey’s test.
Answer: We had revised the paragraph as suggested. Please see the line 310-314 in revised manuscript.
Lines 342-354: Improve.
Answer: We had revised the paragraph in the line 319-326 in revised manuscript.
Lines 356-360: Improve.
A possible suggestion:
Figure 7. Response to different stresses of Coniella vitis strain GP1 (WT), GP1 empty vector transformant (CK) and two CvpalF silenced strains (CvpalFi-17 and CvpalFi-29) growth for 7 days (28 ℃, in the dark) on Potato Dextrose Agar amended with 1M KCl, NaCl, CaSO4, MgSO4, H2O2 or Congo red: (A) Colony morphologies and (B) Colony diameter. Histograms report the average of nine replicates ± SE, different letters indicate significant differences (P < 0.05) according to Tukey’s test.
Answer: We had revised the paragraph as suggested. Please see the line 328-332 in revised manuscript.
Lines 361-380: Improve.
Answer: We had revised the paragraph in the line 335-349 in revised manuscript.
Lines 382-385: Improve.
Explain the significance of sections A, B and C.
A possible suggestion:
Figure 8. Coniella vitis strain GP1 (WT), and two CvpalF silenced strains (CvpalFi-17 and CvpalFi-29) gene expression: A) genes related to Pal signaling pathway, B) plant cell wall degrading enzymes, and C) ????? . Relative expression was normalized using actin as an internal control. The relative gene expression levels in WT were taken as the value of 1. The primers used are listed in Table S1.
Answer: We had revised the paragraph as suggested. Please see the line 350-355 in revised manuscript.
Discussion
Lines 394-399: use “In this study, a homologous gene of PalF/Rim8 was identified in C. vitis. The CvpalF silenced strains CvpalFi-17 and CvpalFi-29 were also obtained using an RNAi approach with the fungal transformation vector pSilent-1 through PEG-mediated protoplast transformation. The applied method was successful in both Ascomycota and Basidiomycota [37–39]. Furthermore, the role of CvpalF in growth, virulence, and response to the different pH conditions was elucidated. Compared with C. vitis WT and CK, CvpalF silenced strains reduce linear growth, mycelia biomass production and worse pathogenicity.” instead of “In this study … Figure 5C).”.
Answer: We had revised the paragraph as suggested. Please see the line 364-370 in revised manuscript.
Line 410: use “Previous” instead of “Our previous”.
Answer: We had used “Previous” instead of “Our previous” in the line 380 in revised manuscript.
Lines 413-422: Improve.
Lines 428-436: Improve.
Answer: We had revised the paragraph in the line 381-408 in revised manuscript.
Line 343: delete (ROS). The abbreviation was not used.
Answer: We had deleted “(ROS)” in the line 403 in revised manuscript.
Lines 460-472: Improve.
Answer: We had revised the paragraph in the line 430-439 in revised manuscript.
Line 484: use “contained” instead of “c.ontained”.
Answer: We had changed “contained” to “c.ontained” in the line 451 in revised manuscript.